# Rpd3/CoRest-mediated activity-dependent transcription regulates the flexibility in memory updating in *Drosophila*

Mai Takakura[1], Reiko Nakagawa [2], Takeshi Ota[3], Yoko Kimura[1], Man Yung NG[4], Abdalla G. Alia[4], Hiroyuki Okuno[5] & Yukinori Hirano[1,6✉]

Consolidated memory can be preserved or updated depending on the environmental change. Although such conflicting regulation may happen during memory updating, the flexibility of memory updating may have already been determined in the initial memory consolidation process. Here, we explored the gating mechanism for activity-dependent transcription in memory consolidation, which is unexpectedly linked to the later memory updating in *Drosophila*. Through proteomic analysis, we discovered that the compositional change in the transcriptional repressor, which contains the histone deacetylase Rpd3 and CoRest, acts as the gating mechanism that opens and closes the time window for activity-dependent transcription. Opening the gate through the compositional change in Rpd3/CoRest is required for memory consolidation, but closing the gate through Rpd3/CoRest is significant to limit future memory updating. Our data indicate that the flexibility of memory updating is determined through the initial activity-dependent transcription, providing a mechanism involved in defining memory state.

[1] Hakubi Center, Kyoto University Graduate School of Medicine, Konoecho, Yoshida, Sakyo-ku, Kyoto, Kyoto 606-8315, Japan. [2] Laboratory for Phyloinformatics, RIKEN Center for Biosystems Dynamics Research, Kobe 650-0047, Japan. [3] Shionogi & Co., Ltd, Laboratory for Innovative Therapy Research, Drug Discovery Technologies 1, Shionogi Pharmaceutical Research Center, 3-1-1, Futaba-cho, Toyonaka-shi, Osaka 561-0825, Japan. [4] Division of Life Science, The Hong Kong University of Science and Technology, Clear Water Bay, Kowloon, Hong Kong, China. [5] Department of Biochemistry and Molecular Biology, Graduate School of Medical and Dental Sciences, Kagoshima University, Sakuragaoka, Kagoshima 890-8544, Japan. [6] Present address: Division of Life Science, The Hong Kong University of Science and Technology, Clear Water Bay, Kowloon, Hong Kong, China. ✉email: yukinori@ust.hk

The consolidated form of memory is not completely fixed, but depending on environmental changes, animals can either update or preserve their consolidated memory. This conflicting nature of memory has been linked to behavioral flexibility, the dysfunction of which is related to the autism spectrum disorder[1,2]. Although previous studies have revealed the molecular and neuronal mechanisms that are activated when memory is updated[3–5], it may be possible that the initial memory consolidation event has already determined the flexibility of memory for later updating. However, such observations or mechanisms have not been reported.

A large body of evidence supports that memory consolidation is mediated by activity-dependent transcription[6,7], with an increased interest in assessing these because of their possible links to various types of human cognitive disorders, including autism spectrum disorder, intellectual disability, Alzheimer's disease, and posttraumatic stress disorder[8,9]. Activity-dependent transcription is induced by constitutively expressed transcriptional activators, including CREB ($Ca^{2+}$/cAMP-responsive element-binding protein)[10–12] and its associating protein, CBP (CREB-binding protein), known as a histone acetyltransferase (HAT)[13,14]. Upon neural activation, calcium-dependent pathways activate those transcriptional factors through posttranslational modifications[15–17], resulting in the transcription of so-called immediate-early genes (IEGs). In general, activity-dependent mRNA expression is immediately shut off[18–21], which could be explained by temporal gating mechanism that opens and closes the time window for activity-dependent transcription. The transcriptional shutoff can be possibly mediated by the transcriptional repressors, such as histone deacetylase 2 (HDAC2)[22], methyl-CpG binding protein 2 (ref. [23]), aryl hydrocarbon receptor nuclear translocator 2 (ref. [24]), and the nucleosome remodeling and deacetylase (NuRD) complex[21]. Importantly, the NuRD complex has been reported to be involved in shutting off activity-dependent transcription in the mouse cerebellum, by depositing a histone variant, H2A.z, at the promoter of the transcribed genes[21]. The mutant mice lacking a component of the NuRD complex showed defects in dendrite pruning and sensorimotor neural coding[21], indicating the significance of the shutoff of activity-dependent transcription. Although the NuRD complex is involved in the shutoff, it remains unknown which protein(s) actually determine(s) the timing of the opening and closing of the gate for activity-dependent transcription. Therefore, the precise underlying molecular substrates of the gating mechanism remain unknown. The gating mechanism may be related to the balance between homeostatic maintenance and plastic change, and thus could be involved in the flexibility in later memory updating. A key step to this end is to scrutinize and manipulate molecular underpinnings of the gating mechanism underlying activity-dependent transcription.

Activity-dependent transcription and its mechanisms are evolutionally well conserved across species. In flies, an olfactory aversive training paradigm[25–28] has been used to demonstrate activity-dependent transcription in neurons in the memory center mushroom body (MB) and their related neurons[20,29–31]. Similar to *HDAC2* knockout which enhances fear memory[22], the knockdown of *HDAC2* homolog, reduced potassium dependency 3 (*Rpd3*), results in enhanced memory in courtship conditioning[32]. Thus, HDAC2 is one of the well-conserved transcriptional repressors acting as a memory suppressor, the activity of which may be related to the gating mechanism underlying transient activity-dependent transcription. In this study, we sought to understand the gating mechanism underlying transient activity-dependent transcription. For this purpose, we carried out the label-free quantification analysis of Rpd3-interacting proteins in MB, in combination with thermogenetic and optogenetic approaches. We found that the Rpd3/CoRest transcriptional repressor complex is dissociated by neural activation. Rpd3/CoRest dissociation was mediated by the binding of the N-terminal truncated variant of CoRest to Rpd3. This compositional change was regulated by acetylation via CBP, and deacetylation via Rpd3, which had a significant role in the gating for activity-dependent transcription. In vivo, dysfunction in Rpd3/CoRest did not impair memory consolidation, but instead, increased flexibility in memory updating. Thus, our study elucidates the gating mechanism underlying transient activity-dependent transcription, which is significant to define the flexibility in the later memory updating.

## Results

**Neural activity-dependent change in the Rpd3/CoRest complex.** We sought to identify activity-dependent changes in Rpd3-interacting proteins, which could possibly serve as the gating mechanism underlying transient activity-dependent transcription. To this end, we performed an interactome analysis for Rpd3 proteins from MB neurons. Rpd3 was tandemly tagged with FLAG and HA, and expressed via the MB247-switch (MBsw) driver[33], expression of which is induced in MB neurons by feeding flies food containing RU486 (RU). We thermogenetically activated most MB neurons by expressing the thermo-sensitive cation channel dTRPA1 (ref. [34]), instead of using the normal olfactory training paradigm, which activates only a subset of MB neurons (5–10%)[35,36]. This thermogenetic manipulation enabled us to handle thousands of flies, in which MB neurons were homogenously activated. The activation of MB neurons was confirmed by the phosphorylation of extracellular signal-related kinase (pERK)[18,20,37], a neural activation marker (Fig. 1a). We then purified the tagged Rpd3 proteins from MB neurons via tandem-tag affinity purification using approximately 2000 flies, with or without thermogenetic activation for 1 h, in order to fully capture the molecular changes (Fig. 1b). The purified immunocomplex was analyzed by a shotgun liquid chromatography-mass spectrometry (LC-MS/MS) analysis to identify the proteins interacting with Rpd3. As a negative control, the flies without dTRPA1 expression were similarly analyzed in order to prevent any effects induced by heat shock. HDAC2 forms three distinct complexes, notably Sin3A, NuRD, and CoRest complexes[38]. We found the amounts of the peptides derived from Mi-2, a component of the NuRD complex, and CoRest, were relatively abundant in the Rpd3-immunoconplex after thermogenetic activation (Supplementary Fig. 1a and Supplementary Table 1). Although other proteins were also found in the Rpd3 immunocomplex, in this study we focused on these known and conserved associating proteins.

Consistent with a phenotype in HDAC2 knockout mice[22], *Rpd3* knockdown enhanced memory formation after a single aversive olfactory training (Supplementary Fig. 1b), which does not normally induce memory consolidation to long-term memory (LTM)[25], via RNAi induced in MB neurons (Supplementary Fig. 1c). If CoRest or Mi-2 are involved in Rpd3 function for memory, their knockdown should also result in memory enhancement. Indeed, memory 1 day after a single training was enhanced by MB-specific knockdown of *CoRest* (Supplementary Fig. 1d), via RNAi targeted to the N-terminal region of *CoRest* (Fig. 1c and Supplementary Fig. 1e). Cycloheximide-feeding impaired memory enhancement by knockdown of *Rpd3* or *CoRest* (Supplementary Fig. 1f, g), suggesting that the enhanced memory is derived from LTM mediated by de novo gene expression. Knockdown of *Mi-2* did not affect memory 1 day after a single training (Supplementary Fig. 1b). These results support the idea that Rpd3 function for memory is mediated by CoRest.

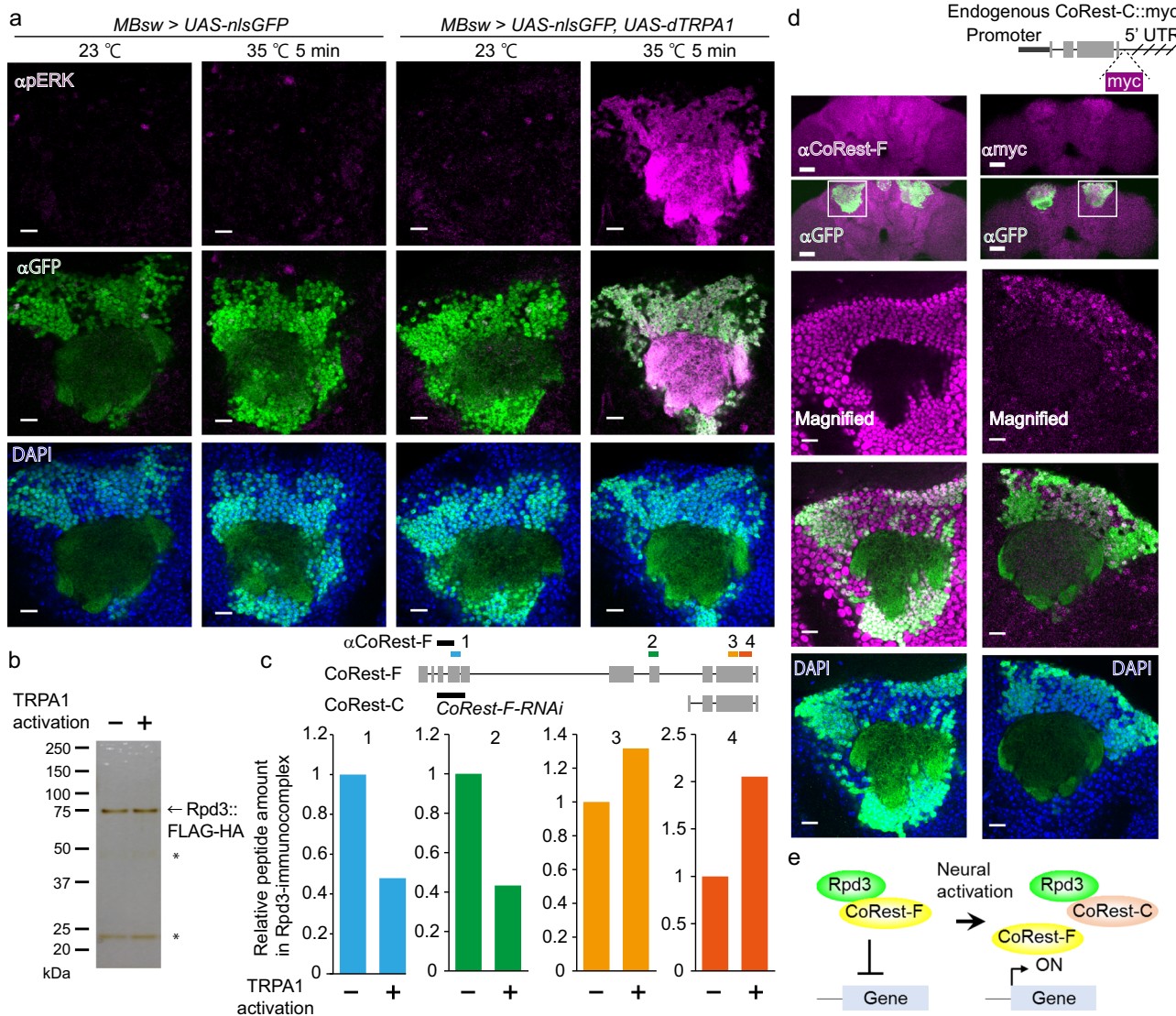

**Fig. 1 Interactome analysis of Rpd3 in MB neurons. a** Thermogenetic activation of MB neurons. GFP fused to the nuclear localization signal (nlsGFP) and dTRPA1 was induced in MB neurons using MBsw. The brains were immunostained with anti-GFP (green) and anti-pERK (magenta) antibodies, and DAPI (blue). The images are representative of experimental replicates ($n = 3$, 4, 4, and 4). Scale bar: 10 μm. **b** Purified Rpd3 proteins from MB neurons. Rpd3:: FLAG-HA was expressed by MBsw, together with dTRPA1. The flies were heat-shocked at 35 °C for 1 h, and the heads were used for tandem-tag affinity purification. The proteins were visualized by silver staining. Asterisks indicate the IgG heavy and light chains. **c** The amount of the representative CoRest peptides identified in the LC-MS/MS analysis. See more details in Supplementary Table 2. (Top) The vertical gray bars indicate the exons, and the horizontal lines indicate the region of the introns. The regions targeted by the anti-CoRest antibody and RNAi were indicated as black bars. **d** Expression of CoRest-F and CoRest-C. (Left) The flies expressing nlsGFP by MBsw were used to immunostain CoRest-F with the anti-CoRest antibody which detected the N-terminal domain of CoRest (magenta). (Right) The genomic region of CoRest-C was cloned, tagged with myc, and inserted into a different region of the genome. Flies also expressing nlsGFP by MBsw were immunostained with the anti-myc antibody (magenta). The images are representative of three experimental replicates. (Upper two panels) Scale bar: 10 μm. (Lower three panels) Magnified view depicted as white squares. Scale bar: 2 μm. **e** Working hypothesis of the complex compositional change in Rpd3/CoRest. Source data are provided as a Source Data file.

Intriguingly, CoRest-binding to Rpd3 was altered in an isoform-specific manner. The isoforms of CoRest contain the full length (CoRest-F) and the N-terminus truncated form (CoRest-C) (FlyBase, http://flybase.bio.indiana.edu/, Fig. 1c). The amount of peptide derived from CoRest-F in the Rpd3 immunocomplex was reduced by thermogenetic activation, whereas those common in both CoRest-F and CoRest-C were increased by thermogenetic activation (Fig. 1c). To confirm that CoRest-C is expressed in MBs, the genomic region containing CoRest-C was cloned, tagged by myc, and inserted at the different genomic loci. Immunostaining with an anti-myc antibody demonstrated that CoRest-C was preferentially expressed in MB

neurons (Fig. 1d), suggesting that CoRest-C is transcriptionally active in MB neurons from an intronic promoter. CoRest-F was broadly expressed in the brain (Fig. 1d), which was detected by an antibody against the N-terminal domain of CoRest-F (Fig. 1c). Taken together, in MB neurons, the Rpd3/CoRest complex could undergo the compositional change after neural activation (Fig. 1e).

**CoRest-C is required for the dissociation of Rpd3 from CoRest-F.** We next confirmed the compositional changes in the Rpd3/CoRest complex in MB neurons. For precise temporal

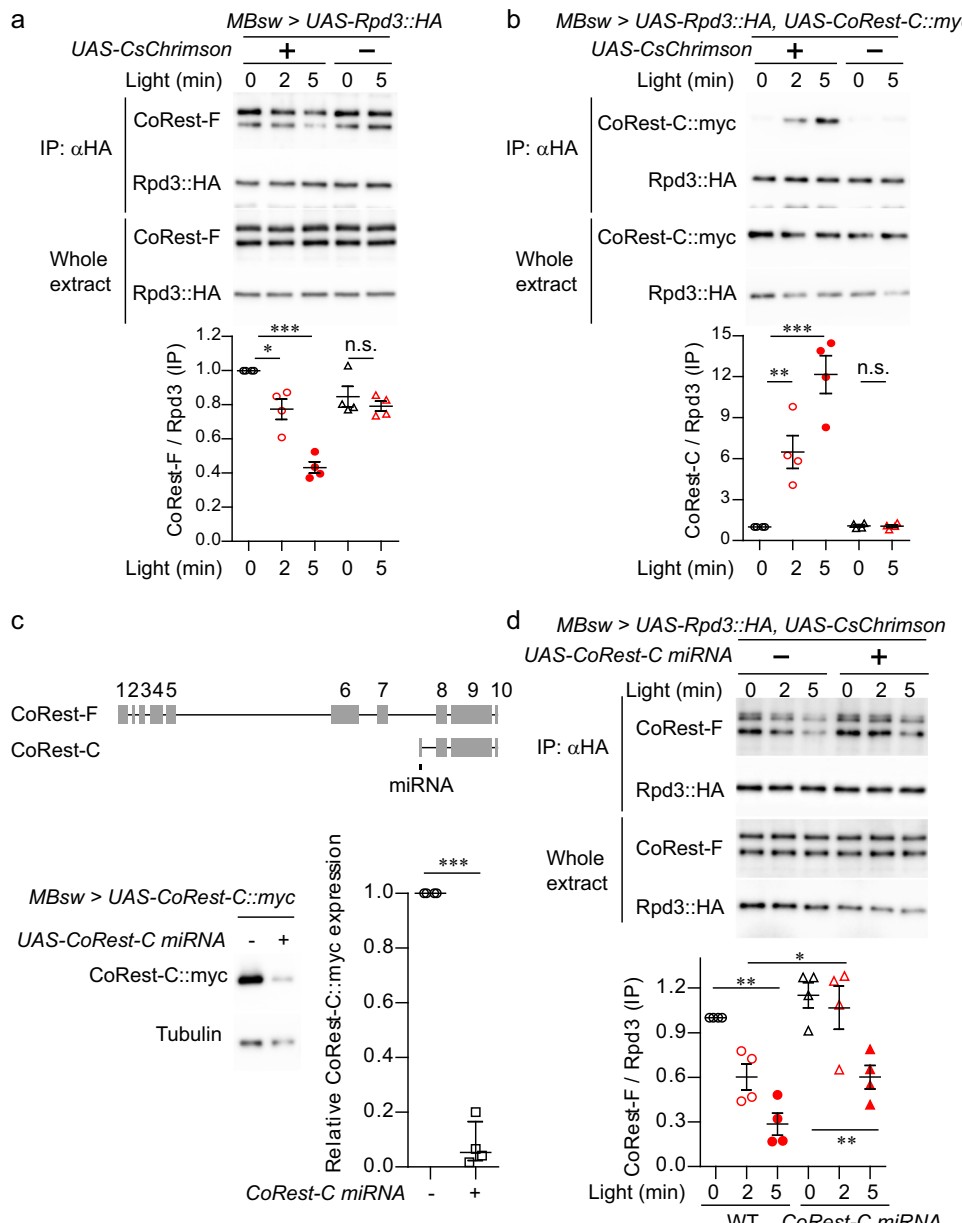

**Fig. 2 CoRest-C is required for the dissociation of Rpd3/CoRest-F. a, b, d** CoRest-F or CoRest-C binding to Rpd3 after neural activation. Flies with the indicated transgenes were fed RU, and illuminated with red light, and the head extracts were immunoprecipitated with the anti-HA antibody. CoRest-F was detected with the anti-CoRest antibody (**a, d**), and other proteins were detected with antibodies specific to the indicated epitope tags in a western blot analysis. One-way ANOVA ($P < 0.0001$; $n = 4$ for all) followed by Tukey's multiple comparisons (two sided) were performed. **c** *CoRest-C* was knocked down with miRNA targeted to the specific exon in CoRest-C. The vertical gray bars indicate the exons, and the horizontal lines indicate the region of the introns. The flies carrying the indicated transgenes were fed RU for 3 days, and the head extracts were analyzed by a western blot analysis with anti-myc and anti-tubulin antibodies. The amount of tubulin was used for normalization. Two-sided Mann–Whitney $U$-test, $P = 0.0002$; $n = 4$. Data are represented as mean ± s.e.m. n.s., not significant, $P > 0.05$; *$P < 0.05$; **$P < 0.01$; ***$P < 0.001$, which is based on experimental replicates using different pooled samples, and the number is indicated as $n$ above. Source data are provided as a Source Data file.

control of neural activation, the red-shifted channelrhodopsin, CsChrimson[39] was expressed by MBsw, together with HA-tagged Rpd3, and MB neurons were optogenetically activated by red light illumination (Supplementary Fig. 2). Coimmunoprecipitated CoRest-F with Rpd3-HA were detected as two bands, which corresponding to known CoRest-F isoforms (for details see "Methods"), when using an antibody against the N-terminal domain of CoRest-F (Fig. 2a, c). We found that CoRest-F interacted with Rpd3 at the basal state, and importantly, this interaction was decreased after optogenetic activation (Fig. 2a). On the other hand, myc-tagged CoRest-C expressed by MBsw was

coimmunoprecipitated with Rpd3-HA only after optogenetic activation (Fig. 2b). These complex changes induced by optogenetic activation were reversed within 15 min (Supplementary Fig. 3a, b), which may provide support for the transient time window of activity-dependent transcription. To test the causal link between CoRest-C and the dissociation of Rpd3/CoRest-F, *CoRest-C* was knocked down by miRNA targeted to a specific exon in *CoRest-C* which is absent from *CoRest-F* (Fig. 2c). Rpd3/CoRest-F dissociation was attenuated by the expression of *CoRest-C*-miRNA (Fig. 2d), suggesting that CoRest-C is critical to the dissociation. We also found that CoRest-C binding to

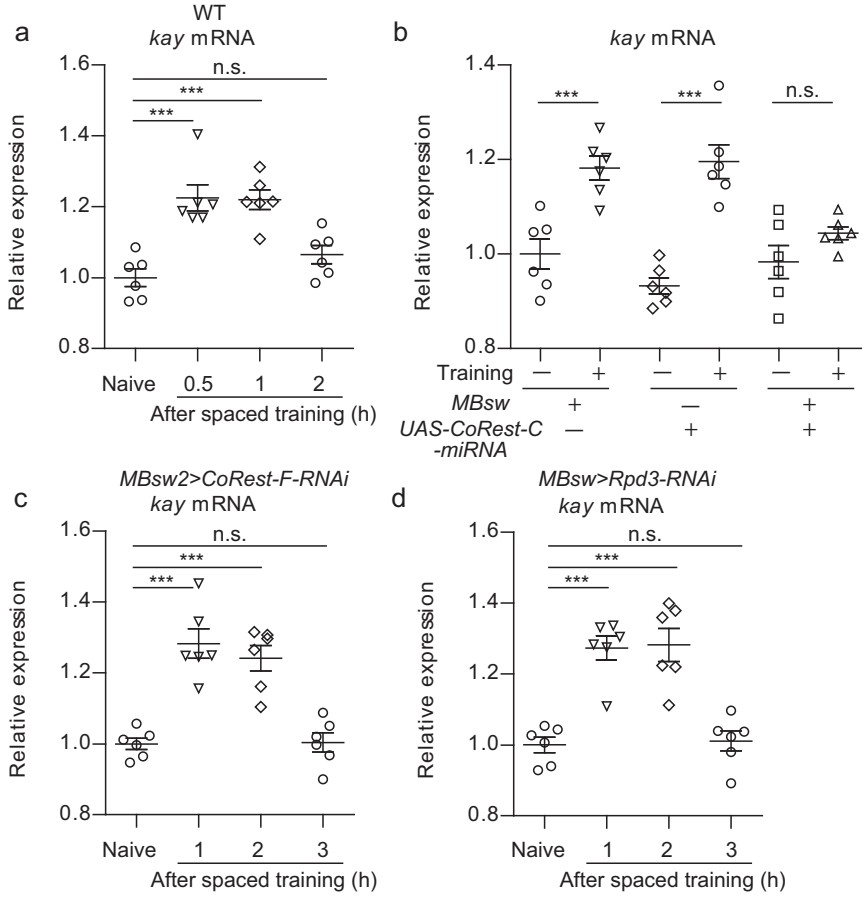

**Fig. 3 The expression of *kay* mRNA is regulated by Rpd3/CoRest. a, b** The expression of *kay* mRNA after spaced training requires CoRest-C. The flies carrying the indicated transgenes were fed normal food (**a**) or food containing RU (**b**), and subjected to spaced training. RNA extracted from the fly heads at the indicated time (**a**) or 1 h (**b**) after spaced training were analyzed via RT-qPCR. **c, d** Dysfunction of Rpd3/CoRest-F delays the shutoff of *kay* mRNA expression. The flies were analyzed as in **b**. The genetic controls are shown in Supplementary Fig. 4. A one-way ANOVA ($P < 0.0001$; $n = 6$ for all) followed by Tukey's multiple comparisons (two sided) was performed. Data are represented as mean ± s.e.m. n.s., not significant, $P > 0.05$; ***$P < 0.001$, which is based on experimental replicates using different pooled samples, and the number is indicated as *n* above. Source data are provided as a Source Data file.

Rpd3 was increased by spaced training, as well as massed training (Supplementary Fig. 3c) that induces anesthesia-resistant memory, another long-lasting form of memory that is independent of gene expression. These results suggest that neural activation induces the compositional change in the Rpd3/CoRest complex.

**Rpd3/CoRest is involved in the gating mechanism underlying transient activity-dependent transcription.** To investigate the role of Rpd3/CoRest in activity-dependent transcription, we examined the mRNA expression of one of the IEGs, *kay*, which is the *Drosophila* homolog of *Fos*. Flies were subjected to spaced training, a repeated aversive olfactory training paradigm with rest intervals, which induces memory consolidation into LTM via gene expression[25]. Increase in *kay* mRNA expression in the heads was increased at 30 min and 1 h after spaced training, and returned to basal levels at 2 h (Fig. 3a)[18]. *CoRest-C* knockdown, which impaired the dissociation of Rpd3/CoRest-F, inhibited the increase in *kay* mRNA expression after spaced training (Fig. 3b). In contrast, the knockdown of either *CoRest-F* or *Rpd3* delayed shutoff of *kay* mRNA expression (Fig. 3c, d and Supplementary Fig. 4a–d), without affecting the basal expression level of *kay* mRNA (Supplementary Fig. 4e). Thus, CoRest-C is required for learning-dependent *kay* transcription, whereas Rpd3/CoRest-F is involved in the shutoff of *kay* transcription, suggesting that

Rpd3/CoRest-F dissociation mediated by CoRest-C is involved in the gating mechanism underlying transient activity-dependent transcription.

**Posttranslational modification of CoRest-C is important for the complex change.** The compositional change in the Rpd3 complex may be induced by posttranslational modifications of either CoRest-F, Rpd3, or CoRest-C. To determine which components undergo such modifications responsible for the complex change, we performed a pull-down assay, in which the binding of fly head proteins to the recombinant proteins was assessed by immunoprecipitation. If CoRest-F is modified after neural activation, inducing the dissociation from Rpd3, CoRest-F expressed in MB neurons after neural activation should not bind to the recombinant Rpd3. However, CoRest-F expressed in MB neurons was pulled down by the recombinant Rpd3 proteins even after optogenetic activation, suggesting that there is no modification to CoRest-F responsible for the dissociation of Rpd3/CoRest-F (Fig. 4a). In contrast, Rpd3 expressed in MB neurons showed reduced binding to recombinant CoRest-F after optogenetic activation (Fig. 4b). This reduced binding was abolished by the knockdown of *CoRest-C* (Fig. 4b), supporting the idea that CoRest-C is important for the dissociation of Rpd3/CoRest-F. We hypothesized that CoRest-C may be modified by neural activation, after which the modified CoRest-C may bind to Rpd3,

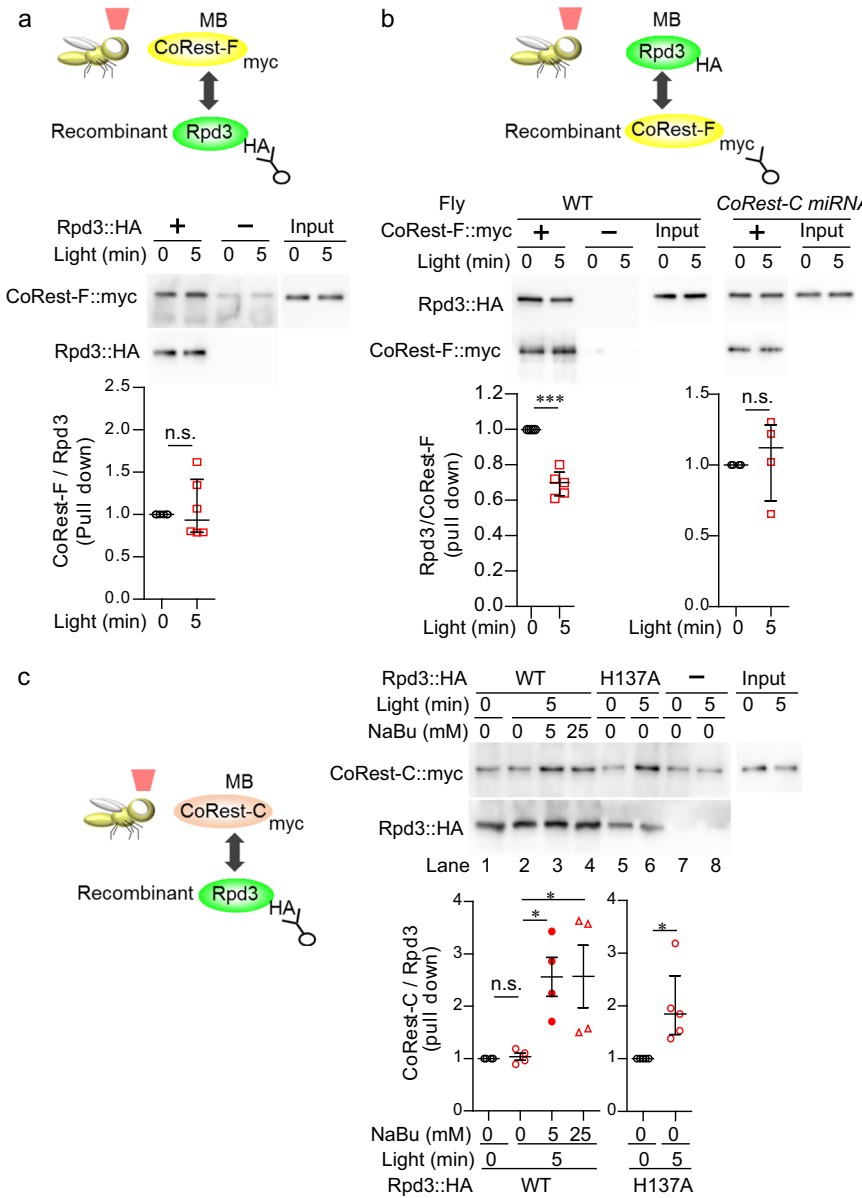

**Fig. 4 Neural activation induces a molecular change in CoRest-C. a** Binding of CoRest-F proteins to the recombinant Rpd3 proteins was not altered by optogenetic activation. The flies carrying MBsw, *UAS-CsChrimson*, and *UAS-CoRest-F::myc* were fed RU, and illuminated with red light. The head extracts were pulled down with the recombinant Rpd3::HA, which was expressed in Sf9 insect cells. Two-sided Mann–Whitney *U*-test, $P = 0.6475$; $n = 6$. **b** Binding of Rpd3 proteins to the recombinant CoRest-F proteins was reduced by optogenetic activation, depending on CoRest-C. Rpd3::HA and CsChrimson were expressed by MBsw in WT flies or in flies carrying *UAS-CoRest-C-miRNA*, and the flies were illuminated with red light. The head extracts were pulled down with recombinant CoRest-F::myc, which was expressed in *E. coli*. (Left) Two-sided Mann–Whitney *U*-test, $P = 0.0007$, $n = 5$. (Right) Two-sided Mann–Whitney *U*-test, $P = 0.7481$, $n = 4$. **c** Binding of CoRest-C proteins to the recombinant Rpd3 proteins was increased by optogenetic activation, when the deacetylation activity of Rpd3 was inhibited. CoRest-C::myc and CsChrimson were expressed by MBsw, and the flies were illuminated with red light. The head extracts were pulled down with recombinant Rpd3::HA (WT proteins or proteins carrying H137A mutation), which was expressed in Sf9 insect cells. The indicated concentration of the HDAC inhibitor sodium butyrate (NaBu) was included in the reaction. (Left) One-way ANOVA ($P = 0.0078$; $n = 6$) followed by Tukey's multiple comparisons (two sided) was performed. (Right) Two-sided Mann–Whitney *U*-test, $P = 0.0369$; $n = 6$. The antibodies for the indicated epitope tags were used in a western blot analysis. Data are represented as mean ± s.e.m. n.s., not significant, $P > 0.05$; *$P < 0.05$; ***$P < 0.001$, which is based on experimental replicates using different pooled samples, and the number is indicated as $n$ above. Source data are provided as a Source Data file.

thereby inhibiting Rpd3/CoRest-F binding. However, in contrast to our expectations, CoRest-C expressed in the MBs did not bind to the recombinant Rpd3 proteins after optogenetic activation (Fig. 4c, lanes 1 and 2). We then speculated that if CoRest-C is modified by acetylation, once it binds to Rpd3, Rpd3 will deacetylate CoRest-C, and cause its dissociation from Rpd3. We therefore added the HDAC inhibitor NaBu to the reaction buffer,

and found that CoRest-C binding to Rpd3 was increased in the presence of NaBu (Fig. 4c, lanes 3 and 4). We further confirmed that CoRest-C bound to the mutant Rpd3 (Rpd3-H137A), which has a point mutation in the catalytically important histidine residue in the HDAC domain[40,41] in the absence of NaBu after optogenetic activation (Fig. 4c, lanes 5 and 6). These results indicate that the acetylation of CoRest-C is important for binding

to Rpd3, and raised the possibility that the complex change in Rpd3/CoRest is mediated by CoRest-C acetylation.

**Acetylation of CoRest-C mediates the dissociation of the Rpd3/CoRest-F complex.** If the acetylation of CoRest-C is important for binding to Rpd3, we should be able to detect acetylated CoRest-C on chromatin in which Rpd3 acts as the repressor. We thus assessed the subcellular localization and the acetylation status of CoRest-C. Nuclear extracts were separated into chromatin and nuclear-soluble fractions by adding MgCl₂, which precipitates the chromatin together with the associating proteins[42] (Fig. 5a). Optogenetic activation induced the chromatin localization of CoRest-C (Fig. 5a, "input"). To assess CoRest-C acetylation, these fractions were subjected to immunoprecipitation with an anti-acetyl lysine antibody. CoRest-C was found in the immunocomplex of the chromatin-bound fraction after optogenetic activation (Fig. 5a, "IP:αAc"), indicating that the chromatin-localized CoRest-C is acetylated. Next, acetylation sites of CoRest-C were analyzed via LS-MS/MS analysis (Fig. 5b), and were identified at K36, K303, and K318 (Supplementary Fig. 5). We then asked which lysine residues are important for the binding of CoRest-C to Rpd3. While the substitution of K36 and K303 to R (K36/303R mutation) did not affect the binding of CoRest-C to Rpd3, the K318R mutation decreased it, and the triple K/R mutations abolished it (Fig. 5c), suggesting that K318 is the major acetylation site affecting Rpd3 binding. Accordingly, *K318R* or the triple *K/R* mutations significantly attenuated the chromatin localization and acetylation of CoRest-C (Fig. 5d). We thus generated *K318R* mutation knock-in flies using CRISPR/Cas9. The *K318R* mutation attenuated the dissociation of the Rpd3/CoRest-F complex (Fig. 5e), suggesting that K318 acetylation is important for the dissociation of the Rpd3/CoRest-F complex. We found that CoRest-F was also acetylated by optogenetic activation (Supplementary Fig. 6a). However, the *K318R* mutation has no effect on CoRest-F-binding to Rpd3 (Fig. 5e), suggesting that CoRest-F does not require acetylation for its binding to Rpd3.

We further determined the acetyltransferase involved in the acetylation of CoRest-C. Among acetyltransferases, we previously reported that CBP was the only one required for LTM formation[43]. Consistently, knockdown of *CBP* attenuated acetylation (Fig. 5f), chromatin localization (Fig. 5f), and binding to Rpd3 (Supplementary Fig. 6b) of CoRest-C, as well as dissociation of the Rpd3/CoRest-F complex (Supplementary Fig. 6c). In contrast, *Rpd3* knockdown delayed the deacetylation of CoRest-C after optogenetic activation (Fig. 5g), suggesting that the acetylation and deacetylation of CoRest-C are mediated by CBP and Rpd3, respectively. We also noted that *Rpd3* knockdown attenuated the chromatin localization of CoRest-C, and that acetylated CoRest-C was observed in the nuclear-soluble fraction (Fig. 5g), suggesting that Rpd3 serves as a CoRest-C binding platform on chromatin.

**CBP, CoRest-C, and Rpd3 colocalize to specific gene loci.** Our findings suggest that the proteins involved in the acetylation of CoRest-C should colocalize at specific gene loci in MB neurons. To focus on MB neurons, we applied our previously reported method to collect MB nuclei[43]. The MB nuclear envelope was labeled with FLAG-KASH, which is inserted into the outer membrane of the nuclear envelope. The MB nuclei were then collected through immunoprecipitation with the anti-FLAG antibody (Fig. 6a), and these nuclei were subjected to ChIP-seq analysis for CoRest-C, Rpd3, and CBP, with or without optogenetic activation (Fig. 6b). Binding of all proteins was enriched near the transcriptional start site (Fig. 6c–e). There were 1237

binding sites identified for CoRest-C, 2717 for Rpd3, and 6684 for CBP (Fig. 6f and Supplementary Tables 3–5). Importantly, among CoRest-C-binding sites, there were 1038/1237 (83.9%) overlapping sites between CoRest-C and CBP, 789/1237 (63.8%) between CoRest-C and Rpd3, and 704/1237 (56.9%) between CoRest-C, Rpd3, and CBP (Fig. 6f and Supplementary Table 6), including the *kay* gene locus (Fig. 6b). Although we may have underestimated the frequency of overlap, due to the threshold of peak calling, the colocalization of CoRest-C, CBP, and Rpd3 supports the model in which CBP-dependent acetylation of CoRest-C induces its binding to Rpd3. Gene ontology (GO) analysis indicated that 704 genes showed the enrichment in the biological process GO terms related to development and morphology (Supplementary Fig. 7a and Supplementary Table 7), and the molecular function GO terms related to DNA-binding transcription factor activity and cytoskeletal protein binding (Supplementary Fig. 7b and Supplementary Table 7).

We further assessed the transcriptional profile after optogenetic activation by RNA-seq using the collected MB nuclei (Supplementary Fig. 8a). Nuclei were fixed during the preparation to preserve nuclear RNA, and the RNAs recovered from cross-linking, which were fragmentated due to the treatment at the high temperature (see Online Methods), were collected by oligo-dT beads. The obtained RNA-seq data therefore contains the reads from the 3′ region of each gene, which indicated the mRNA expression level (Supplementary Fig. 8a). The principal component analysis demonstrated the clear segregation of gene expression profiles across time after optogenetic activation (Supplementary Fig. 8b). We found that upregulation of gene expression is robust at 10 min after a 5-min optogenetic activation (Fig. 6g). There were 2702 genes showing significant differences at this time point, of which 1582 genes had increased expression, while the remaining had decreased expression (Fig. 6g). Among the 1582 genes with increased expression, 1055 genes were bound by CBP (Fig. 6h), supporting the idea that CBP is an important factor in activity-dependent transcription. Furthermore, 338 were bound by CoRest-C, 254 of which showed colocalization with CBP and Rpd3, including *kay* (Fig. 6h). Thus, these 254 genes are strong candidates in which transcriptional activation is controlled via CoRest-C and Rpd3. Our nuclear RNA-seq demonstrated that on-going transcription in the nucleus terminated within 15 min (Fig. 6g), which is consistent with the kinetics of an Rpd3/CoRest compositional change (Supplementary Fig. 3a, b).

We then examined the mRNA expression of the representative genes from the list of enriched GO terms (Supplementary Table 7), *Myo31DF* (GO, cytoskeletal protein binding, post-embryonic animal organ development; ortholog of mammalian *Myo1d*), *sr* (GO, DNA-binding transcription factor activity; ortholog of mammalian *Egr2*), *h* (GO, DNA-binding transcription factor activity; ortholog of mammalian *Hes1*), and *grh* (GO, DNA-binding transcription factor activity, sensory organ development; ortholog of mammalian *Grhl1*). The mRNA expression of these genes was transiently upregulated at 1 h and shut off at 2 h after spaced training in WT flies (Supplementary Fig. 9a). Similarly to *kay* mRNA, *CoRest-F* or *Rpd3* knockdown prolonged the expression of these mRNA up to 2 h after spaced training (Supplementary Fig. 9b–e). These data provide a comprehensive list of activity-dependent genes that are regulated by CoRest-C, Rpd3, and CBP.

**Biological significance of Rpd3/CoRest.** Our findings raised the possibility that the transient activity-dependent transcription is tightly regulated through the acetylation of CoRest-C. In support of this idea, knockdown of *CoRest-C* and the *K318R* mutation

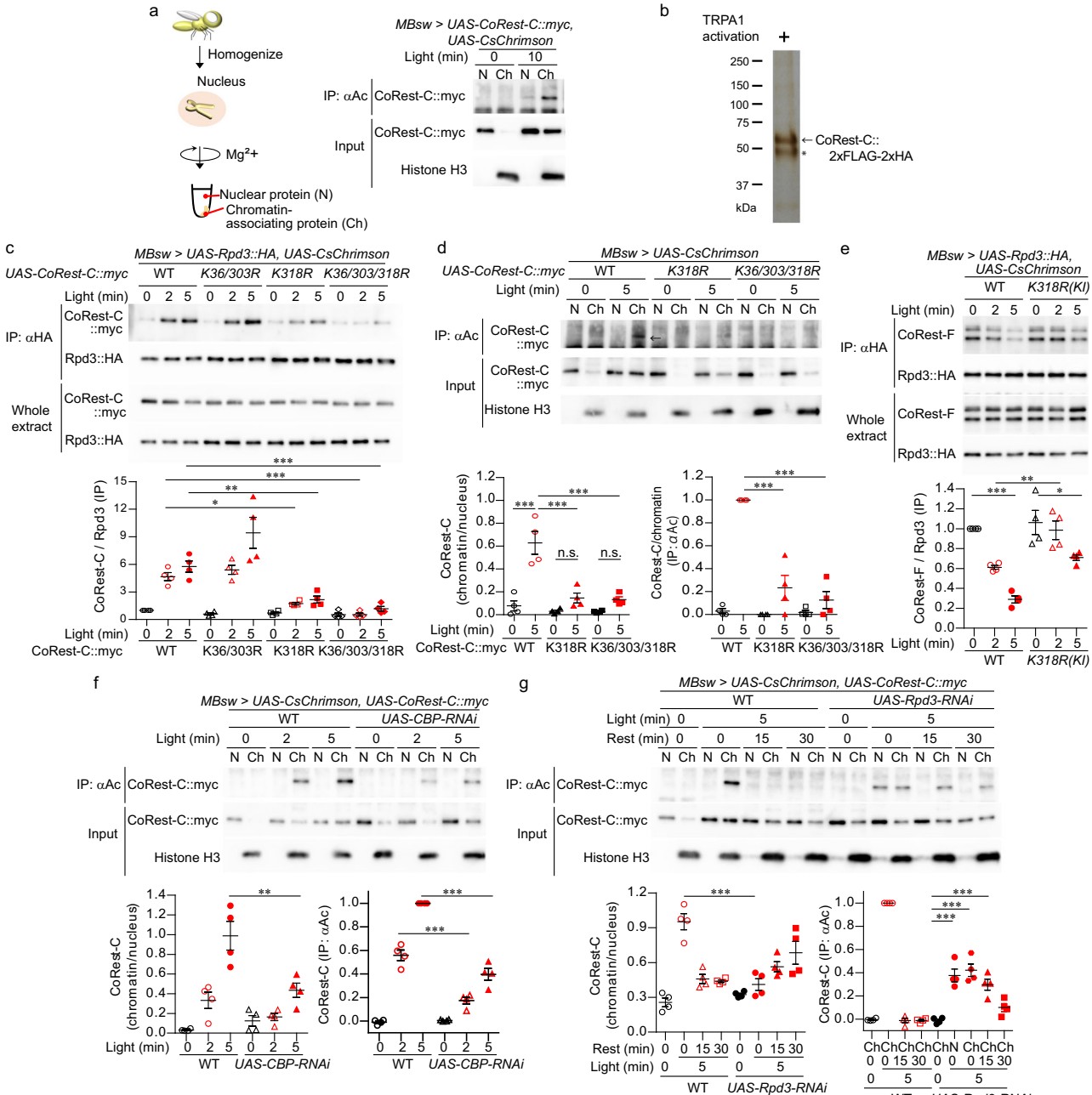

**Fig. 5 Acetylation of CoRest-C is important for binding to Rpd3. a**, **d** Acetylated CoRest-C was localized on chromatin after optogenetic activation. The flies carrying the indicated transgenes were fed RU, and illuminated with red light. Nuclear extracts prepared from the heads were separated into the nuclear proteins (N) and the chromatin-associating proteins (Ch) by centrifugation in the presence of Magnesium (Mg²⁺). Each fraction was immunoprecipitated with an anti-Acetyl lysine antibody (αAc). The antibodies for the indicated epitope tags or for histone H3 were used in a western blot analysis. **a** The images are representative of three experimental replicates. **d** A one-way ANOVA ($P < 0.0001$; $n = 4$ for all) followed by Tukey's multiple comparisons (two sided) was performed. The arrow indicated CoRest-C::myc. **b** Purified CoRest-C proteins from MB neurons. CoRest-C::FLAG-HA was expressed by MBsw, together with dTRPA1. The flies were heat-shocked at 35 °C for 30 min, and the heads were subjected to tandem-tag affinity purification. The proteins were visualized by silver staining. Asterisk indicates the IgGs. **c**, **e** CoRest-C binding to and CoRest-F dissociation from Rpd3 after optogenetic activation requires K318 of CoRest-C. The flies with the indicated transgenes were fed RU, and illuminated with red light, and the head extracts were immunoprecipitated with the anti-HA antibody. CoRest-F was detected with the anti-CoRest antibody (**e**), and other proteins were detected with antibodies specific to the indicated epitope tags in a western blot analysis. *K318R (KI)*, *CoRest-C-K318R* knock-in mutation. A one-way ANOVA ($P < 0.0001$; $n = 4$ for all) followed by Tukey's multiple comparisons (two sided) was performed. **f**, **g** Acetylation and deacetylation of CoRest-C is mediated by CBP and Rpd3. The flies carrying the indicated transgenes were fed RU, and illuminated with red light and analyzed as in **a**. A one-way ANOVA ($P < 0.0001$; $n = 4$ for all) followed by Tukey's multiple comparisons (two sided) was performed. Data are represented as mean ± s.e.m. n.s., not significant, $P > 0.05$; *$P < 0.05$; **$P < 0.01$; ***$P < 0.001$, which is based on experimental replicates using different pooled samples, and the number is indicated as $n$ above. Source data are provided as a Source Data file.

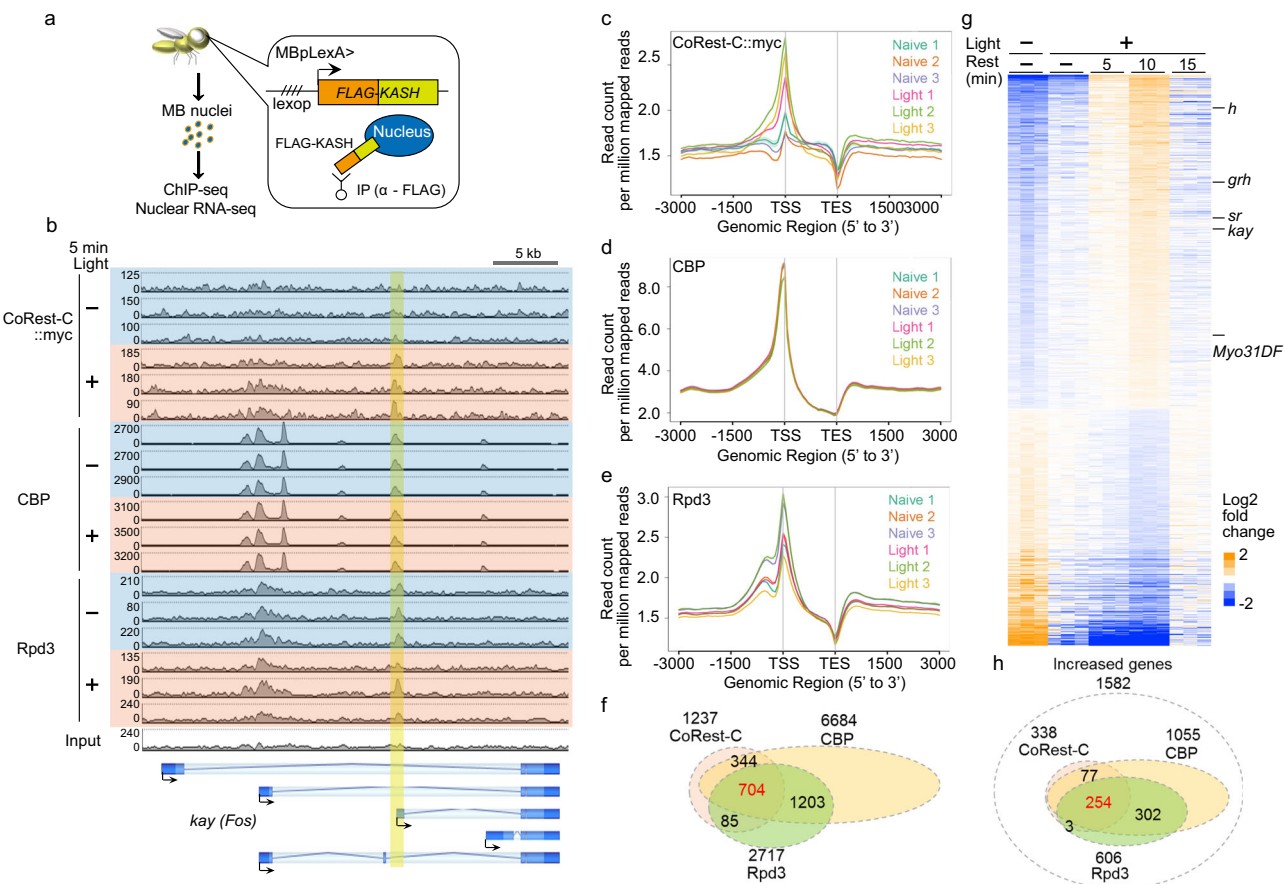

**Fig. 6 CoRest-C, Rpd3, and CBP colocalize to the specific gene loci. a** Schematic diagram of the preparation of MB nuclei. The KASH domain, which is inserted to the outer membrane of the nuclear envelope, was tagged with FLAG, and expressed in MB neurons using the *MBp-LexA* driver[43]. The FLAG-labeled MB nuclei were collected by immunoprecipitation with the anti-FLAG antibody. **b** ChIP-seq signals at the region near *kay*. FLAG-KASH and CsChrimson were expressed by MBpLexA. CoRest-C::myc was expressed by MBsw. MB nuclei prepared from naïve flies or flies illuminated with red light were subjected to ChIP-seq analysis, using anti-myc ($n = 3$), anti-CBP ($n = 3$), or anti-Rpd3 ($n = 3$) antibodies. The sequencing data of input DNA were shown at the bottom. Overlapped binding sites determined by peak calling are highlighted by a yellow vertical bar. The *y* axes show the number of mapped reads, and the upper limits were adjusted to the number of total reads in each sample. **c–e** Aggregate gene plot. The mapped reads of each sample were summarized with respect to the distance from the transcriptional start site (TSS) and the transcriptional end site (TES). **f** Venn diagram showing overlaps in CoRest-C-, CBP-, and Rpd3-binding sites. **g** Heatmap of the log2 fold change in expression of 2702 genes significantly altered after 5 min of optogenetic activation followed by 10 min of rest. Increased genes, 1582; decreased genes, 1120. **h** Venn diagram showing the overlap in CoRest-C-, CBP-, and Rpd3-binding sites in the 1582 genes showing increased expression in **g**.

impaired *kay* mRNA expression after spaced training (Fig. 3b and Supplementary Fig. 10a), as well as *Myo31DF*, *sr*, *h*, and *grh* (Supplementary Fig. 10b–e). Importantly, these genetic manipulations also impaired the olfactory aversive LTM formation (Fig. 7a, b). Knockdown of *CoRest-C* and the *K318R* mutation did not affect memory which does not require gene expression, 1-day memory after massed training (anesthesia-resistant memory, Supplementary Fig. 11a, c), or 1-h memory after a single training (short-term memory; Supplementary Fig. 11b, d). Therefore, CoRest-C is specifically required for activity-dependent transcription and subsequent memory consolidation to LTM.

We next addressed the biological significance of Rpd3/CoRest-F-mediated shutoff of activity-dependent transcription in olfactory aversive spaced training paradigm. However, neither *CoRest-F* nor *Rpd3* knockdown affected 1-day memory after spaced training (Fig. 7c and Supplementary Fig. 12a), suggesting that Rpd3/CoRest-F-mediated shutoff of activity-dependent transcription is involved in processes other than memory consolidation. We then asked whether or not the consolidated aversive memory could be flexibly updated, by subjecting the flies to a reversal

learning paradigm[44], in which an odor-shock association was reversed the following day (Fig. 7d). At the immediate test after reversal learning, the control flies adapted to the reversed association, although this behavior was much weaker than that in response to normal training, due to its conflicting nature with the previous spaced training (Fig. 7d). Interestingly, the *CoRest-F* or *Rpd3* knockdown flies more flexibly adapted to the reversed association than the control flies, suggesting that the flexibility in memory updating is increased by *CoRest-F* or *Rpd3* knockdown (Fig. 7d and Supplementary Fig. 12b). Learning itself was not affected by the knockdown of *CoRest-F* or *Rpd3* (Supplementary Fig. 12c, d). Acute knockdown of *CoRest-F* or *Rpd3* after spaced training did not affect reversal learning (Supplementary Fig. 12e). The increased flexibility in memory updating by *CoRest-F* and *Rpd3* knockdown was specific to reversal learning, and was not due to nonspecific sensory experience, since the effects of *CoRest-F* and *Rpd3* knockdown were not observed in memory after exposure to the conditioned odors or unpaired training (Supplementary Fig. 12e–h). These data indicate that Rpd3/CoRest-F is involved in limiting flexibility in memory updating.

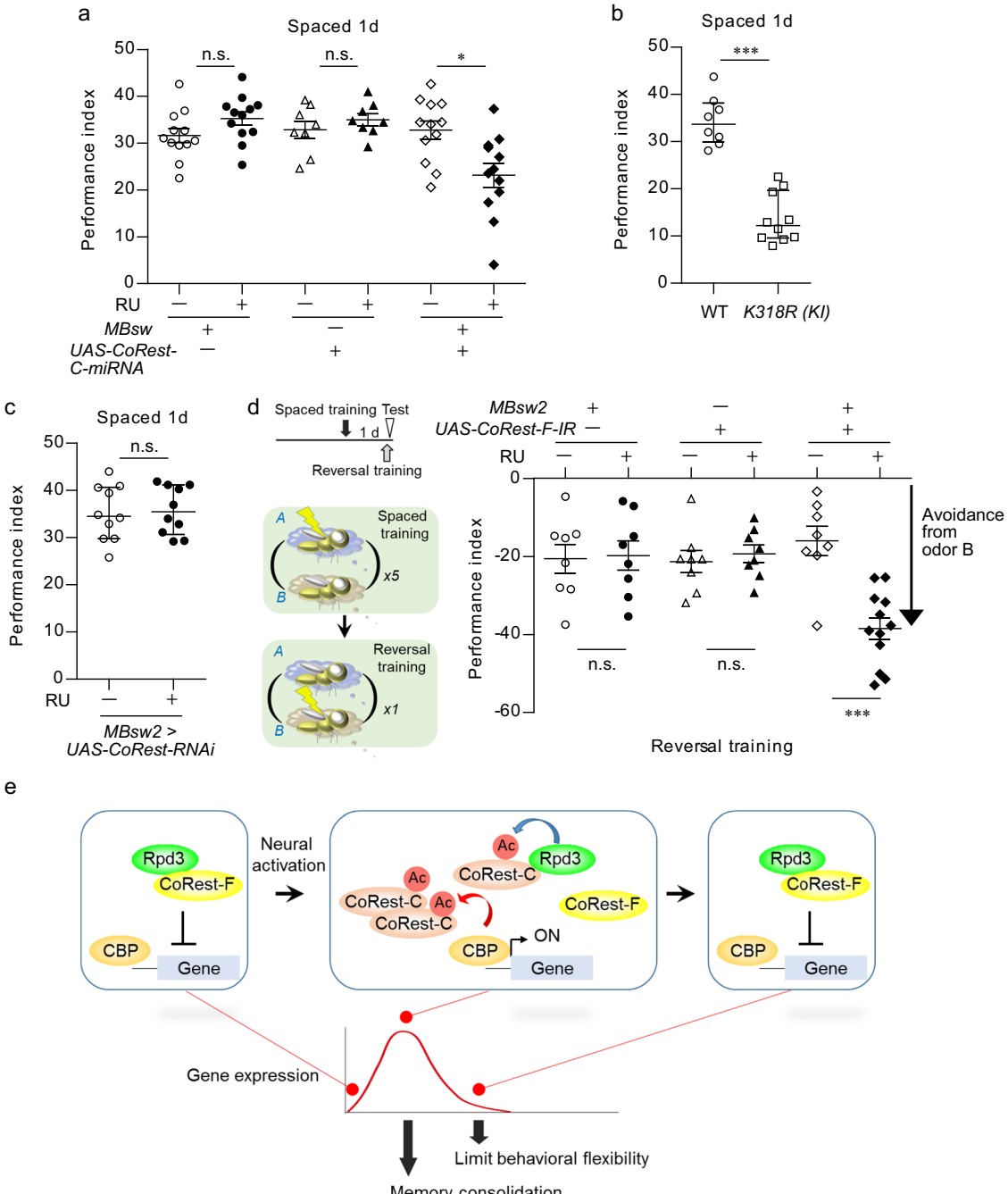

**Fig. 7 CoRest-C is required for memory consolidation, whereas CoRest-F is required to limit behavioral flexibility. a**, **b** Dysfunction in *CoRest-C* impaired memory consolidation. The flies with indicated transgenes were fed RU in **a**, or normal food in **b**, and 1-day memory after spaced training was analyzed. **a** A Kruskal–Wallis test ($P = 0.0027$; $n = 12$) followed by Dunn's multiple comparison test (two sided) was performed. **b** Two-sided Mann–Whitney $U$-test, $P <$ 0.0001; $n = 8$–10. *K318R (KI)*, *CoRest-C-K318R* knock-in mutant. **c** Knockdown of *CoRest-F* did not affect memory consolidation. The flies with indicated transgenes were fed RU, and 1-day memory after spaced training was assessed. Two-sided Mann–Whitney $U$-test, $P = 0.9118$; $n = 10$. **d** The flexibility in memory updating was increased by *CoRest-F* knockdown. The flies with indicated transgenes were fed RU, and subjected to spaced training. On the following day, the flies were subjected to reversal learning, in which the association of the odors and shock were alternated, and memory was immediately measured. A Kruskal–Wallis test ($P = 0.0006$; $n = 12$) followed by Dunn's multiple comparison test (two sided) was performed. **e** Model. Acetylated CoRest-C via CBP, binds to Rpd3 to destabilize the Rpd3/CoRest-F complex. This reaction enables activity-dependent transcription, but is recovered by the deacetylation of CoRest-C by Rpd3. Activity-dependent transcription via the destabilization of Rpd3/CoRest-F is important for memory consolidation, whereas its shutoff by Rpd3/CoRest-F limits the flexibility of memory updating. Data are represented as mean ± s.e.m. n.s., not significant, $P > 0.05$; *$P < 0.05$; ***$P < 0.001$, which is based on experimental replicates using different pooled samples, which number is indicated as $n$ above. Source data are provided as a Source Data file.

## Discussion

In this study, through an interactome analysis of Rpd3 in MB neurons, we provided evidence pointing to the temporal gating mechanism underlying activity-dependent transcription (Fig. 7e). Neural activation induces acetylation of CoRest-C via CBP, which destabilizes the transcriptional repressor complex Rpd3/CoRest-F, enabling activity-dependent transcription for memory consolidation. Given that acetylation of CoRest-C is reversed by Rpd3, resulting in its dissociation from Rpd3, the acetylation of CoRest-C could act as a temporal gating mechanism for activity-dependent transcription. Interestingly, we found that when the function of Rpd3/CoRest-F is impaired, memory consolidation remains intact, but the resulting memory is more flexibly updated by subsequent reversal learning. This suggests that opening and closing the temporal gate for activity-dependent transcription through Rpd3/CoRest is important for memory consolidation and behavioral flexibility, respectively, and proposes the model in which the molecular mechanism acting in memory consolidation could affect the later memory updating by controlling the memory state.

**The downstream event related to the flexibility in memory updating**. The observation that Rpd3/CoRest-F dysfunction increases the flexibility in memory updating is derived from the immediate test after reversal learning, which excludes the effect of transcription after reversal learning. Thus, the flexibility in memory is already determined prior to reversal learning, which is presumably enabled via activity-dependent transcription following initial training. IEGs in mammals include many genes encoding transcription factors, such as c-Fos, Egr1, Egr2, and Npas4 (ref. [6]). Consistently, the genes bound by CoRest-C, CBP, and Rpd3 include transcription-related genes (Fig. S7). This supports the previous model that the first wave of transcription leads to a second wave of transcription, which will be important in neural functioning[6,45]. Although we have not determined whether the second wave of transcription is also affected by knockdown of *Rpd3* or *CoRest-F*, future research should assess both waves in order to further elucidate the causal association between transcription and behavioral flexibility. The recent study demonstrated that olfactory aversive spaced training paradigm using two odors, one odor associated with electric shock and the other not, produces two complementary memories, an aversive memory and a safe-memory, respectively[46]. It will be intriguing to know which memory is affected by reversal learning, and how the activity-dependent transcription fits to the physiological aspects in memory updating.

**The biological significance of the shutoff of activity-dependent transcription**. We found that dysfunction in Rpd3/CoRest-F delayed the shutoff of activity-dependent transcription, and resulted in higher flexibility in memory updating, without affecting memory consolidation per se. HDAC2 forms three separate complexes, the CoRest complex, Sin3 complex, and NuRD complex[38]. The NuRD complex is involved in the shutoff of transcription in the cerebellum in mice; however, in contrast to our finding, it is required for a form of classical conditioning, eyeblink conditioning[21], suggesting that the NuRD-dependent shutoff of transcription is involved in memory formation in mouse cerebellum. Another previous report indicated that inhibition of HDAC2 does not impair memory, but enhances memory in fear conditioning in mice[22]. These seemingly contradictory results may be explained by the different target genes of the three HDAC2 complexes. In addition, recent studies have revealed that the activity-dependent genes differ across cell types[47,48]. Therefore, alternatively, the aforementioned

discrepancy may be due to differences in the targeted neurons. Anyhow, our data provide a model describing the behavioral significance of temporal gating for activity-dependent transcription, alongside elucidating its molecular details.

**The temporal gating mechanism underlying activity-dependent transcription**. Our results suggest that the acetylation of CoRest-C drives activity-dependent transcription by binding to Rpd3, which destabilizes the Rpd3/CoRest-F complex. The tight regulation of the CoRest-C acetylation, through CBP and Rpd3, can be considered as a gate for activity-dependent transcription. We believe that any biochemical events which switch activity on and off may serve as a temporal gating mechanism, with the acetylation of CoRest-C being one of them. For instance, the acetylation of CoRest-C via CBP is regulated by neural activity, perhaps via a posttranslational modification of CBP such as phosphorylation[49]. The mechanism underlying the activation and inactivation of CBP may also act as a temporal gating mechanism. It is important to note that, although recombinant Rpd3 does not bind to CoRest-C unless deacetylase activity is blocked (Fig. 4c), endogenous Rpd3 binds to CoRest-C in vivo (Fig. 2b). This suggests that the deacetylation activity of Rpd3 is attenuated by neural activation through a posttranslational modification or by binding to unknown molecules. If this is the case, again, the regulation of Rpd3 deacetylase activity may also be one of the temporal gating mechanisms. Nonetheless, given that Rpd3/CoRest-F is the terminal effector in transcriptional repression, and CoRest-C directly regulates the stability of Rpd3/CoRest-F complex, the acetylation of CoRest-C may be a central gating switch for activity-dependent transcription. We note that approximately one-fifth of activity-dependent upregulated genes are bound by CoRest-C (338/1582 genes), suggesting that alternative mechanisms may act on other gene loci.

While a greater degree of flexibility in memory updating may be advantageous for animals to adjust their behavior, it may fail to preserve the important information inherent to past memory. These conflicting requirements for memory updating and the preservation should be balanced by some molecular mechanism. Here, we have demonstrated that the shutoff of activity-dependent transcription during the initial memory consolidation actively limits the flexibility of the later memory updating. The resulting limited flexibility in memory updating may have been beneficial for flies to preserve consolidated memory. This function, enabled by the Rpd3/CoRest complex, may be evolutionarily conserved in mammalian Rcor3, which is one of the orthologs of *Drosophila* CoRest, and possibly expressed in a full-length isoform and an isoform with only a C-terminal domain (the Ensembl database). Given that the flexibility in memory may be tightly linked to the development of human behavior, our study may lay the foundation for an additional model of autism spectrum disorder or intellectual disability, linked to the shutoff of activity-dependent transcription.

## Methods

**Ethics statement**. The study design was approved by the appropriate ethics review board by Kyoto University.

**Fly stocks and culture conditions**. The RNAi lines were obtained as follows: the UAS-Rpd3-IR (30600), UAS-CBP-IR (105115), and UAS-Mi-2-IR (107204) lines from the Vienna *Drosophila* RNAi Center (Vienna, Austria); the UAS-CoRest-IR line (3878R-3) from the National Institute of Genetics (Shizuoka, Japan). The *MBsw* line[33] was obtained from R. Davis; and the UAS-nlsGFP, UAS-IVS-CsChrimson.mVenus[39], LexAop-IVS-CsChrimson.mVenus, UAS-dTRPA1 (ref. [34]), UAS-IVS-myr::tdTomato, vas-Cas9 (ref. [50]) lines from Bloomington *Drosophila* Stock Center (Indiana, USA), and the attp2 line from Kyoto DGGR (130390, Kyoto, Japan)[51]. The lexop-FLAG-KASH and MBp-LexA lines were previously described[43]. The *MBsw2* flies were generated, which eliminate the leaky expression of *MBsw*, since CoRest-RNAi induced by original *MBsw* was effective even in

uninduced condition (Supplementary Fig. 13; see below for details of plasmid construction). The *CoRest-C::myc* knock-in, *MBsw2*, *UAS-Rpd3::FLAG-HA*, *UAS-CoRest-C::myc* (WT or K/R mutant), *UAS-CoRest-C-miRNA*, and *UAS-CoRest-C:: FLAG-HA* lines were obtained via germline transformation using standard procedures. The *CoRest-K318R* mutant flies was obtained via germline transmission using *vas-Cas9* flies by injecting pCFD4-CoRest-2xgRNA and the donor plasmid, pCoRest::K318R (see below for details regarding plasmid construction). The *CoRest-K318R* mutation was confirmed by sequencing genomic DNA fragments amplified via polymerase chain reaction (PCR). Fly lines used in this study were outcrossed with our wild-type control line, w(CS10)[52], for at least five generations before use.

Flies were raised under a 12-h light:dark cycle, at a temperature of 25 °C and humidity of 60%. All experiments were performed during the light cycle. Flies carrying MBsw were raised at 20 °C to reduce the possibility of leaky expression. RU486 (RU; Mifepristone, Sigma, St. Louis, MO, USA) was dissolved in ethanol and mixed with fly food, to a final concentration of 0.5 mM RU. All-*trans*-retinal (Sigma, St. Louis, MO, USA) was dissolved in ethanol and mixed with fly food, at a final concentration of 0.4 mM all-*trans*-retinal. The flies were fed these food for 3 days prior to each experiment. For cycloheximide (CHX)-feeding, Whatmann 3MM filter paper was soaked with a 5% sucrose solution containing 35 mM CHX and 0.5 mM RU, and flies were reared on the paper for 1 day.

**Plasmid constructs**. To construct the UAS-Rpd3::FLAG-HA plasmid, oligonucleotides carrying the sequence of 3xGlycine linker (3xGGGS)-fused 2xFLAG-2xHA were used for PCR amplification of the coding region of Rpd3 from cDNA, which was cloned into the *Eco*RI–*Xba*I-digested pUAST vector[53], resulting in pUAS-Rpd3-2xFLAG-2xHA. The UAS-CoRest-C::FLAG-HA was obtained similarly. To construct the pUAST-CoRest-C::myc plasmid, oligonucleotides carrying the sequence of 3xGlycine linker (3xGGGS) was used to amplify 9xmyc fragments via PCR using pYM6 carrying 9xmyc[54], which was cloned into the *Xho*I–*Xa*I-digested pUAST vector, resulting in pUAST-9myc. The coding region of CoRest-C was amplified by PCR from cDNA, and cloned into the *Bgl*II–*Xho*I-digested pUAST-9myc vector, resulting in pUAST-CoRest-C::myc. The pUAST-CoRest-F:: myc plasmid was similarly obtained. The pUAST-CoRest-C::myc plasmids carrying the K36/303R, K318R, or K36/303/318R (triple K/R) mutation were generated by introducing substitution mutations by PCR. The CoRest-C::myc plasmid used to detect endogenous expression of CoRest-C (Fig. 1d) was obtained by cloning the genomic region of CoRest-C, including 2 kb of 5′UTR and 1 kb of 3′UTR, with 9xmyc insertion at the C terminus, into the *Sph*I–*Pst*I-digested pUAST, which excluded the UAS sequence.

The pET28a-CoRest-F::myc plasmid, which was used to express the recombinant CoRest-F protein in *Escherichia coli* (Fig. 4b), was obtained by cloning the CoRest-F::myc fragment amplified by PCR from the pUAST-CoRest-F::myc plasmid, into the *Bam*HI–*Hin*dIII-digested pET28a (Sigma, St. Louis, MO, USA). The pFastBac-Rpd3::FLAG-HA plasmid, which was used to express the recombinant Rpd3 protein in Sf9 insect cells (Fig. 4a, c), was obtained by cloning the Rpd3-2xFLAG-2xHA fragment amplified by PCR from the pUAST-Rpd3-2xFLAG-2xHA plasmid, into the *Spe*I–*Xho*I-digested pFastBac (Thermo Fisher Scientific, San Jose, CA, USA). The pFastBac-Rpd3::FLAG-HA plasmid carrying the H137A mutation was generated by introducing substitution mutations by PCR.

The pWALIUM20-CoRest-C plasmid, which was used to induce miRNA-based knockdown of CoRest-C, was obtained by cloning oligonucleotides carrying the sequence of the specific exon of CoRest-C into the *Nhe*I–*Eco*RI-digested pWALIUM20 vector (Drosophila Genomics Resource Center, IN, USA). To obtain the pMBsw2, firstly a DNA fragment containing 247 bp of the sequence from the *Mef2* gene (which is an enhancer active in MB neurons) and the *hsp70Bb* minimal promoter was amplified by PCR from the pMBpLexA-lexop-FLAG-KASH plasmid[43], which was cloned to the *Sph*I–*Bgl*II-digested pUAST, eliminating the UAS sequence, resulting in pMBp. Oligonucleotides carrying the sequence of LexA hinge region (a.a. 71–98)[55] was used to amplify the DNA-binding domain of Gal4 (a.a. 1–64) by PCR from the pBPGAL4.2Uw-2 plasmid (Addgene no. 26227). The progesterone receptor ligand-binding domain fused to p65 activation domain was amplified by PCR from pSwitch #1 (Drosophila Genomics Resource Center, IN, USA). These two fragments were cloned into the *Bgl*II–*Xho*I-digested pMBp plasmid, resulting in pMBsw2 (Supplementary Fig. 13).

To construct the pCFD4-CoRest-C-K318R-2xgRNAs, two gRNA target sequences were added to the fragment amplified from pCFD4 (Addgene no. 49411), as previously described[50]. The PCR fragment was cloned into the *Bbs*I-digested pCFD4, resulting in pCFD4-CoRest-C-K318R-2xgRNAs. The sequences of gRNAs targeting nearby the K318 of *CoRest-C* were 5′-gagcgcgatttttcttcgccgg-3′ and 5′-gcggctgctcggcgacgggc-3′. To construct the donor plasmid, pCoRest-C-K318R, the 1.5 kb genomic region containing CoRest-K318 was amplified, with introduction of K318R mutation. Substitutions of PAM sequences without changing amino acid sequence were included. This 1.5 kb fragment was cloned into the *Kpn*I–*Sac*I-digested pBluescript SK-(−) vector, resulting in the donor plasmid.

**Thermogenetic and optogenetic manipulation**. For thermogenetic neural activation, the vials containing the dTRPA1-expressing flies were transferred to the incubator at 35 °C, and the flies were immediately frozen in the liquid nitrogen at the indicated time. Optogenetic neural activation was induced to the CsChrimson-

expressing flies, which were fed 0.4 mM all-*trans*-retinal for 3 days, and the flies were frozen in the liquid nitrogen after red light illumination. Light illumination was performed in the clear plastic vials surrounded by four red-emitting light-emitting diodes (LEDs) (617 nm) (SP-03-E4; Luxeon StarLEDs, Brantford, ON, Canada) as previously described[18]. Pulsed illumination was applied at 40 Hz with a pulse-width of 12.5 ms.

**Shotgun LC-MS/MS analysis**. The analyzed proteins were prepared by tandem-tag affinity purification using approximately 2000 fly heads. The head extracts were prepared in extraction buffer (20 mM Tris-HCl at pH 8.0, 100 mM NaCl, 0.1% *n*-dodecyl-β-D-maltoside (DDM), 1 mM EDTA, Complete Protease Inhibitor Cocktail (Sigma, St. Louis, MO, USA), 1 mM PMSF, and 5 mM sodium butylate), which were homogenized and sonicated. The extracts were immunoprecipitated with ANTI-FLAG M2 Affinity Gel (Sigma, St. Louis, MO, USA) for 2 h at 4 °C, and the beads were washed three times. The bound proteins were released in the extraction buffer using 150 μg/mL of 3xFLAG peptides (F4799, Sigma, St. Louis, MO, USA), and further immunoprecipitated with anti-HA antibody-conjugated magnetic beads (88837; Thermo Fisher Scientific, San Jose, CA, USA) for 2 h at 4 °C. The bound proteins were eluted by incubating in the urea buffer (100 mM TEAB at pH 8.5, 8 M urea, 0.1% Rapigest SF) for 30 min at RT.

Peptide sample preparation and nano flow-liquid chromatography tandem mass spectrometry was performed as described previously[56] with some modification. In brief, the eluted proteins were digested with 10 μg/mL modified trypsin (Sequencing grade, Promega), and for posttranslational analysis for acetylation, the CoRest IP fractions were subjected to in-gel digestion with 10 μg/ mL modified trypsin (Sequencing grade, Promega) and Glu-C Protease (MS Grade, Thermo Scientific) at 37 °C for 16 h. The digested peptides were desalted with in-house made C18 Stage-tips, dried under a vacuum, and dissolved in 2% acetonitrile and 0.1% formic acid. The peptides mixtures were then fractionated by C18 reverse-phase chromatography (3 μm, ID 0.075 mm × 150 mm, CERI). The peptides were eluted at a flow rate of 300 nL/min with a linear gradient of 5–35% solvent B over 60 min.

The raw files were searched against the *Drosophila melanogaster* dataset (Uniprot Proteome Drosophila Melanogaster 2017.02 downloaded) with the common Repository of Adventitious Proteins (cRAP, ftp://ftp.thegpm.org/fasta/ cRAP) using the MASCOT version 2.6 (Matrix Science) via Proteome discoverer 2.1. with a false discovery rate (FDR) set at 0.01. Carbamidomethylation of cysteine, oxidation of methionine, and acetylation of protein N-termini were set as fixed modification. For identification of lysine acetylation site, both acetylation (42.0106) and tri-methylation (42.0470) of lysine were included as variable modifications. Number of missed cleavages site was set as 2. The measurement of the amount of the identified peptides, including CoRest, was performed with skyline[57]. The amount of immunoprecipitated Rpd3 was determined by Proteome discoverer 2.1. The relative amount of the peptides were then determined (Fig. 1c and Supplementary Table 1) by dividing each amount of the peptides in the TRPA1-activated samples by those in the untreated samples, whose value is further normalized by the amount of Rpd3.

**Immunoprecipitation**. The head extracts from 50 to 100 flies were prepared as above, and immunoprecipitated with anti-HA antibody-conjugated magnetic beads (88837; Thermo Fisher Scientific, San Jose, CA, USA) for 2 h at 4 °C. The beads were washed three times with extraction buffer, and the eluted proteins were analyzed by western blot analysis.

In pull-down assay, the recombinant CoRest-F::myc proteins expressed in *E. coli* cell were immunoprecipitated with anti-myc antibody-conjugated magnetic beads (88843; Thermo Fisher Scientific, San Jose, CA, USA), in extraction buffer for 2 h at 4 °C. The recombinant Rpd3::HA proteins expressed in Sf9 cells were immunoprecipitated with anti-HA antibody-conjugated magnetic beads in phosphate-buffered saline containing 0.01% Triton X-100 supplemented with Complete Protease Inhibitor Cocktail, 1 mM dithiothreitol, and 1 mM PMSF for 2 h at 4 °C. The beads bound by CoRest-F::myc or Rpd3::HA were washed three times, and mixed with the fly extracts. The fly extracts were prepared as above in extraction buffer with or without sodium butylate. After 2-h rotation at room temperature, the beads were washed three times with extraction buffer, and the eluted proteins were analyzed by western blot analysis.

**Separation of the chromatin-associating proteins**. The nuclei from heads were prepared using a Teflon/glass homogenizer in extraction buffer without DDM and EDTA. The homogenate was filtered through 100 μm nylon mesh to remove cuticles, and the nuclei were precipitated by centrifugation at 2.3 krfc for 1 min. After the cytosolic proteins in the supernatant were removed, the nuclei were dissolved in extraction buffer without EDTA, containing 10 mM MgCl₂ to precipitate the chromatin and 0.1% DDM to break the nuclear envelop. The chromatin fraction was separated from the nuclear-soluble fraction by centrifugation at 20 krfc for 5 min, and the chromatin-associating proteins were solubilized from the pellets by sonication, in extraction buffer. Each fraction was subjected directly to western blot analysis, or to immunoprecipitation with 20 μL of Protein A/G Agarose (Thermo Fisher Scientific, San Jose, CA, USA) and 4 μg anti-acetyl lysine

antibody (ab21623; Abcam, Cambridge, MA, USA), followed by washes for three times and western blot analysis.

**Generation of antibody**. The anti-Rpd3 and anti-CBP antibodies were raised against the C-terminal 124 amino acids of Rpd3 and the N-terminal 1000 amino acids of CBP, respectively, by Japan lamb (Hukuyama, Hiroshima, Japan). Sera were collected and affinity-purified using a resin conjugated with the antigens. The anti-CoRest antibody raised against the amino acids 634–820 was provided by G. Mandel[58]. The antibodies used in western blot analysis were rabbit anti-CoRest antibody at a 2000-fold dilution; mouse monoclonal anti-myc antibody (626802; BioLegend, San Diego, CA, USA) at a 500-fold dilution; rabbit monoclonal anti-HA antibody (3724; Cell Signaling Technology, Beverly, MA, USA) at a 1000-fold dilution; mouse monoclonal anti-HA antibody (901515; BioLegend, San Diego, CA, USA) at a 1000-fold dilution; rabbit anti-histone H3 antibody (ab1791; Abcam, Cambridge, MA, USA) at a 4000-fold dilution, and mouse anti-α-tubulin antibody (T6199; Sigma, St. Louis, MO, USA) at a 20,000-fold dilution.

**Quantification of the images of western blot**. Images were quantified by Image J (NIH, USA). In the coimmunoprecipitation and pull-down experiments, the band intensities of the coimmunoprecipitated proteins (e.g. CoRest-F) were divided by those of immunoprecipitated proteins (e.g. Rpd3). For CoRest quantification, CoRest-F appeared as two bands in western blotting, which correspond to the full-length isoform and the isoform without exons 6 and 7 (FlyBase, http://flybase.bio.indiana.edu/). Because the two bands of CoRest-F behave similarly, only those of the lower band are measured. In the chromatin fractionation experiments, the band intensities of the chromatin-associating CoRest-C was divided by those of the nuclear-soluble CoRest-C (Fig. 5d, f, g). The relative amount of acetylated CoRest-C was determined as the band intensity of each sample divided by that of WT after optogenetic activation (Fig. 5d, f, g).

**Quantification of transcripts (RT-qPCR)**. Total RNA was extracted from 30 heads using TRIzol reagent (Thermo Fisher Scientific, San Jose, CA, USA), and 125 μg of RNA was used to synthesize cDNA using ReverTra Ace qPCR RT Master Mix with gDNA Remover (TOYOBO, Osaka, Japan). The cDNAs were then analyzed via quantitative real-time PCR (Bio-Rad Laboratories, Hercules, CA, USA). Transcripts of rp49 were used for normalization. The sequences of the primers are listed in Supplementary Table 10.

**Immunohistochemistry**. Staining of pERK was performed as previously described[18] using rabbit monoclonal anti-pERK antibody (4376; Cell Signaling Technology, Beverly, MA, USA) at a 100-fold dilution. Other staining experiments were also performed as previously described[18]. The following primary antibodies were used: rabbit anti-CoRest antibody at a 100-fold dilution; rabbit monoclonal anti-myc antibody (2278; Cell Signaling Technology, Beverly, MA, USA) at a 100-fold dilution; mouse nc82 antibody (Developmental Studies Hybridoma Bank, Univ. Iowa, USA) for the detection of the presynaptic protein Bruchpilot at a 50-fold dilution; rabbit anti-DsRed antibody (632496; Takara, Shiga, Japan) at a 200-fold dilution; and chicken anti-GFP antibody (ab13970; Abcam, Cambridge, MA, USA) at a 2,000-fold dilution. The following secondary antibodies were used at 500-fold dilutions: donkey anti-chicken immunoglobulin Y (IgY) Alexa 488 antibody (703-545-155; Jackson ImmunoResearch Labs, Inc., West Grove, PA, USA), donkey anti-mouse IgG Alexa Fluor 488 antibody (715-545-150; Jackson ImmunoResearch Labs, Inc., West Grove, PA, USA), and donkey anti-rabbit IgG Cy3 antibody (711-166-152; Jackson ImmunoResearch Labs, Inc., West Grove, PA, USA). Images were captured using a confocal microscope LSM780 (Zeiss Microsystems, Jena, Germany) or FV1000 (Olympus, Tokyo, Japan).

**Purification of MB nuclei**. Approximately 500 flies expressing FLAG-KASH in the MBs were used for ChIP analysis, as previously described with some modifications[43]. Heads were collected and homogenized in crosslinking buffer (15 mM Hepes-KOH at pH 7.5, 100 mM NaCl, 0.25 M sucrose, 0.1% NP40, 1 mM dithiothreitol, and 1 mM PMSF) containing 1% formaldehyde, and Complete Protease Inhibitor Cocktail (Sigma, St. Louis, MO, USA) using a Teflon/glass homogenizer. The homogenate was left on ice for 15 min. Crosslinking was quenched by adding 125 mM glycine, and the homogenate was filtered through 40 μm nylon mesh to remove cuticles. The nuclei were rinsed three times in ChIP buffer (20 mM Tris-HCl at pH 8.0, 100 mM NaCl, 1 mM EDTA, 1% Triton X-100, 0.1% SDS) containing 0.25 M sucrose. The nuclei were then dissolved in ChIP buffer, containing 0.25 M sucrose, Complete Protease Inhibitor Cocktail and 1 mM PMSF, briefly sonicated to dissociate the individual nuclei, and immunoprecipitated with ANTI-FLAG M2 Affinity Gel (Sigma, St. Louis, MO, USA) for 2 h at 4 °C. The beads were rinsed four times in extraction buffer, with 5 min nutation at 4 °C between washes, and subjected to ChIP analysis.

**ChIP-seq analysis**. The ChIP-seq analysis was performed as previously described[43]. The purified MB nuclei dissolved in ChIP buffer were sonicated using a Q500 Sonicator (QSonica, Newtown, CT, USA), at a power setting of 20%, for a total of 3 min, resulting in fragmentation of DNA at the average length of 400 bp.

The extracts were centrifuged to remove insoluble materials including the M2 Affinity Gel, and the supernatants were used for immunoprecipitation with 10 μL of Protein A/G Agarose (Thermo Fisher Scientific, San Jose, CA, USA) and the following antibodies: 4 μL of rabbit monoclonal anti-myc (2278; Cell Signaling Technology, Beverly, MA, USA), 0.5 μg of anti-CBP, or 3 μg of anti-Rpd3 antibodies. After overnight incubation at 4 °C, the beads were washed four times with high-salt buffer (20 mM Tris-HCl at pH 8.0, 500 mM NaCl, 1 mM EDTA, 1% Triton X-100, 0.1% SDS), with 5 min nutation at 4 °C between washes. The beads were rinsed with 1× TE buffer (10 mM Tris-HCl at pH 8.0, and 1 mM EDTA), and the immunoprecipitates were eluted with 1× TE buffer containing 1% SDS. Crosslinking was reversed by incubation at 65 °C for 6 h. Next, samples were incubated with 0.1 mg/mL Proteinase K (Takara, Shiga, Japan) at 55 °C for 1 h, the DNA was extracted once with phenol:chloroform and once with chloroform, and then ethanol-precipitated. The resulting DNA (ChIP DNA) was used to prepare a library using NEBNext Ultra II DNA Library Prep Kit for Illumina (New England Biolabs Inc., Ipswich, MA, USA), according to the manufacturer's instructions. Paired-end reads were generated for ChIP and input DNA in HiSeq X ten (Illumina, San Diego, CA, USA).

**RNA-seq analysis**. The MB nuclei collected as above were treated with 0.1 mg/mL Proteinase K (Takara, Shiga, Japan) in 1× TE buffer containing 1% SDS at 65 °C for 4 h. The RNA was extracted once with phenol:chloroform and once with chloroform, and then ethanol-precipitated. DNA was digested with DNase, and mRNA was collected using Oligo d(T)25 Magnetic Beads (New England Biolabs Inc., Ipswich, MA, USA), according to the manufacturer's instructions. The obtained mRNA was used to generate a library using NEBNext Ultra II RNA Library Prep Kit for Illumina (New England Biolabs Inc., Ipswich, MA, USA), according to the manufacturer's instructions. Paired-end reads were generated in HiSeq X ten (Illumina, San Diego, CA, USA).

**Bioinformatic analysis**. Read quality was first assessed using FastQC (version 0.11.4) (http://www.bioinformatics.babraham.ac.uk/projects/fastqc/), and proceeded to adaptor trimming using trimmomatic[59], followed by mapping to the Drosophila reference genome, dm6 from UCSC using STAR[60]. The reads with low mapping quality below 8 and the non-primary mapped reads were eliminated. The summary of the mapping rate is shown in Supplementary Table 9. For ChIP-seq, the retained reads (3.6–13.7 million reads) were used for peak calling with MACS[61] on Strand NGS software (Agilent Technologies, Palo Alto, CA, USA), using a default setting except for the following parameters; $10^{-4}$ as a P value cutoff, and 3 as a enrichment factor. The binding sites were determined when the peaks were overlapped in at least two out of three biological replicates from the optogenetically activated samples. Aggregate gene plot was shown by ngs.plot[62] using the same retained reads as those used for peak calling. GO (gene ontology)-enriched groups were determined on WebGestalt (WEB-based GEne SeT AnaLysis Toolkit)[63]. For RNA-seq, the filtered reads (12.2–20.0 million reads) were analyzed with HTSeq-count[64] to obtain numbers of the reads mapped on exons, which was further analyzed on R using DESeq2. The gene counts (the sum of the read counts on exons) were normalized via DESeq2, which were used to generate heatmap using Treeview[65] (Fig. 6g).

**Behavioral assay**. Aversive olfactory conditioning was performed as previously described[29,66,67]. Briefly, approximately 70 flies were placed in a training chamber, where they were exposed to odors and electrical shocks. During single-cycle training, either 3-octanol (OCT) or 4-methylcyclohexanol (MCH) was paired with electrical shocks (60 V, 1.5-s pulses every 5 s) for 1 min, while the remaining odor was not. For testing, flies were placed at a choice point between the two odors for 1.5 min. A performance index (PI) was calculated so that a 50:50 distribution (no memory) yielded a PI of zero, while a 0:100 distribution away from the shock-paired odor yielded a PI of 100. Individual performance indices were calculated as the average of two experiments, in which the shock-paired odor was alternated.

Spaced and massed training were performed using an automated computer system, which controlled both electric shock and odor application to flies. Spaced training consisted of five single-cycle training sessions (5× spaced training), with a 15-min rest interval between each session[68]. Massed training was performed as five single-cycle training sessions without rest intervals. The trained flies were maintained at 17 °C until testing. Memory was manually quantified as described above.

Reversal learning was performed as previously described[44]. The spaced trained flies were subjected to another training with reversed contingency of the CS–US association, in which the odor previously paired with electrical shocks is not paired with shocks, but the other odor previously unpaired with shock is paired with shocks. The flies were tested immediately after reversal learning. In control experiments (Supplementary Fig. 12e–h), the spaced trained flies were subjected to exposure to the conditioned odor and the unconditioned odor for 1 min, or to unpaired training in which electrical shocks (60 V, 1.5-s pulses every 5 s) for 1 min were delivered to the flies, followed by rest interval for 30 s, exposure to the conditioned odor for 1 min, and the unconditioned odor for 1 min. The flies were then immediately tested.

**Statistical analysis**. No statistical calculations were used to predetermine sample sizes. Our sample sizes are similar to those generally used in this field of research. Flies from each cross were randomly assigned into treatment groups, where possible. All samples were numbered and the investigators were blinded. Statistical analyses were performed using Prism version 5.0. For behavioral data analysis, Mann–Whitney $U$-test was used for comparisons between two groups, and Kruskal–Wallis test followed by Dunn's multiple comparison test was used for comparisons among multiple groups. For analyses of qPCR and western blot, Paired $t$-test was used for comparisons between two groups, and one-way ANOVA followed by Tukey's multiple comparisons was used for comparisons among multiple groups. $P$ values <0.05 were regarded as statistically significant. All data are presented as the mean ± s.e.m. Each experiment was successfully reproduced at least two times and performed on multiple days.

**Reporting summary**. Further information on research design is available in the Nature Research Reporting Summary linked to this article.

## Data availability

All data generated or analyzed during this study are included in this published article (and its Supplementary Information Files) or deposited. The GEO accession number for the ChIP-seq and RNA-seq data reported in this paper is GSE150642. All mass spectrometry data have been deposited to ProteomeXchange Consortium via jPOST with the accession number PXD021294 and JPST000948, respectively. Source data are provided as a Source Data File.

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

## Acknowledgements

We thank the members of the Y. Hayashi laboratory for helpful discussions. We would like to thank G. Mandel, R. Davis, and the Drosophila Genomics Resource Center (NIH grant 2P40OD010949) and Drosophila RNAi Screening Center (NIH Grant 5R24OD019847) for materials. This study was supported by JSPS KAKENHI grants (grant numbers: JP17H04984 and JP16H01274 to Y.H.; JP18H05127 and JP19H03328 to H.O.), Kato Memorial Bioscience Foundation, the Sumitomo Foundation, the Suzuken Memorial Foundation, the Novartis Foundation, the Ichiro Kanehara Foundation, Senri Life Science Foundation (to Y.H.), and Shionogi & CO., LTD (to Y.H. and H.O.).

## Author contributions

M.T. performed the protein work. LC-MS/MS analysis was carried out by R.N. with the support of T.O. M.T., Y.K., and Y.H. established the transgenic flies. Y.K. assisted with immunostaining. M.Y.N.G. and A.G.A. assisted with immunoprecipitation. H.O. generated reagents and tools, and wrote the manuscript with Y.H. Y.H. designed the study, performed other analyses, and wrote the manuscript.

## Competing interests

The authors declare no competing interests.
