## [Peer Review File · Nature Communications]

Reviewers' Comments:

Reviewer #1:

Remarks to the Author:

In the manuscript entitled "Controlling the flexibility in memory updating through the gating mechanism for activity-dependent transcription in *Drosophila*", the authors claim that the changes of activity-dependent transcription regulated by Rpd3/CoRest complex determine memory consolidation or updating. The evidence in support of Rpd3/CoRest-mediated mechanism in regulating activity-dependent transcription is strong, however, the causal link between such mechanism and memory performance is weak and should be strengthened.

Major concerns

- 1) To support that Rpd3 and CoRest-F negatively regulate activity-dependent transcription in memory consolidation, it is important to demonstrate that the memory enhancement after a single aversive olfactory training in Rpd3-knockdown and CoRest-F-knockdown flies is indeed LTM rather than other forms such as ASM or ARM. Experiments like CXM feeding and cold-shock treatment are helpful.
- 2) Does the mechanism demonstrated in Figure 7e occur after spaced training but not massed training? It is helpful to perform experiments like Figures 2a and 2b by using spaced or massed training stimuli instead of artificial activation of MB neurons.
- 3) The major conclusion of this work is that the flexibility of memory updating is already determined in the initial memory consolidation. To strengthen this conclusion, it is necessary to verify that the acute knockdown of CoRest or Rpd3 after memory consolidation does not affect reversal learning.

Reviewer #2:

Remarks to the Author:

In this paper, Takakura and colleagues investigate the role of activity-induced acetylation-mediated changes in the CoRest isoform associated with Rpd3 in memory consolidation and subsequent plasticity. The authors use a series of sophisticated genetic manipulations along with biochemistry and genomic analysis to show that neural activity induces a switch from CoRest-F to CoRest-C association with Rpd3. This switch depends on CBP-mediated acetylation of CoRest-C, and is required for the correct kinetics of repression of activity-induced transcription. Disrupting this pathway prolongs gene activation, and specifically changes the plasticity of memory consolidation. Overall, this is an interesting and complete study, and the data are compelling, rigorous, and well-presented.

Major point:

- 1) In Figure 1, the rationale for focusing on known Rpd3 binding partners is understandable, but the mass spec data need a bit more explanation. Ideally, there should be some representation of the entirety of the data in the main text (for example a graph showing which Rpd3 associated proteins increase, or at least which known Rpd3 complex members change). If this is too distracting from the main text, it should be presented as a detailed supplementary figure. Can the authors provide additional explanation about how they selected CoRest from the mass spec data, and whether they think other differences on activation are important? Also please indicate if the experiment was replicated, and if so how the replicates were handled in the data analysis. The authors should also clarify what they mean by "increased binding" of Mi-2 and CoRest—was this statistically determined, or how was a threshold determined, and which of the metrics indicated in Supplementary Table 1 was used?

Minor point

- 1) In Figure 1, DAPI panels are labelled "Nuclues" please correct to "DNA" or "nuclei" or "DAPI"

Reviewer #3:

Remarks to the Author:

In the study by Mai Takakura et al., the authors sought to understand the basis of the gating mechanism which opens and closes the time window for activity dependent transcription, thought to regulate memory consolidation and if necessary, its update. Specifically, the authors aimed to identify the molecular basis of this mechanism. They focused on the well conserved HDAC2 transcriptional repressor, in *Drosophila* Rpd3, whose activity is involved in both the activity dependent transcription and memory. They embarked on a truly heroic journey starting from a proteomic analysis through biochemical assays to the behavioral validation of their findings. As a result, they found that the gating mechanism of transient transcription involves compositional changes in the Rpd3/CoRest transcriptional repressor complex. Specifically, they determined that the Rpd3/CoRest complex is dissociated upon neural activation by binding of the short CoRest isoform, CoRestC. This compositional change is mediated by acetylation via CPB, and deacetylation via Rpd3. The authors validated those findings in vivo using behavioral assay in *Drosophila*. They determined that Rpd3/CoRest complex has no role in consolidation of the aversive olfactory memory but instead mediates increased flexibility in memory update.

Major comment

This is really a very impressive and exciting work both in terms of approaches and findings. The experiments are carefully controlled, and the conclusions are well supported by the data. In this work the *Drosophila* system was used to its best. The combination of various approaches, from proteomics to behavior, provided an unprecedented detailed picture of the molecular mechanism involved in the activity dependent transcription in memory consolidation and its update.

Minor points

- Both the result and discussion sections of the manuscript are well written. However, the introduction is not very clear and needs some work.
- Names of HDAC2, Rpd3, CoRest should not be abbreviated when mentioned for the first time in the text.
- Please explain why kay mRNA is upregulated after ~1.5 hours while Rpd3/CoRest complexes induced by neuronal activation dissociate within 15 minutes?
- Nucleus is misspelled in DAPI panels of Fig.1a and d
- Degree symbol is messed up in legends of Fig.1 and 5.
- Light is misspelled in Fig.5g (top and bottom).

We appreciate the valuable comments from all reviewers, which have helped to substantially strengthen our manuscript. We also appreciate the reviewers' extremely encouraging comments supporting our findings. In this revised manuscript, we have fully addressed the issues raised by the reviewers. Our point-by-point responses (indicated in black) to the reviewers' comments (indicated in blue) are presented below.

The response to Reviewer #1

In the manuscript entitled "Controlling the flexibility in memory updating through the gating mechanism for activity-dependent transcription in *Drosophila*", the authors claim that the changes of activity-dependent transcription regulated by Rpd3/CoRest complex determine memory consolidation or updating. The evidence in support of Rpd3/CoRest-mediated mechanism in regulating activity-dependent transcription is strong, however, the causal link between such mechanism and memory performance is weak and should be strengthened.

Major concerns:

1) To support that Rpd3 and CoRest-F negatively regulate activity-dependent transcription in memory consolidation, it is important to demonstrate that the memory enhancement after a single aversive olfactory training in Rpd3-knockdown and CoRest-F-knockdown flies is indeed LTM rather than other forms such as ASM or ARM. Experiments like CXM feeding and cold-shock treatment are helpful.

We agree with the reviewer's comment. We added data showing that cycloheximide-feeding impairs memory enhancement by *Rpd3*- and *CoRest-F*-knockdown (Supplementary Fig. 1f,g), indicating that the enhanced memory is LTM mediated by *de novo* gene expression.

2) Does the mechanism demonstrated in Figure 7e occur after spaced training but not massed training? It is helpful to perform experiments like Figures 2a and 2b by using spaced or massed training stimuli instead of artificial activation of MB neurons.

This is a significant point to support the physiological relevance of our findings. We examined CoRest-C binding to Rpd3 after spaced or massed training (Supplementary Fig. 3c), and found that both trainings induced CoRest-C binding to Rpd3. Dissociation of CoRest-F from Rpd3 was not shown, because only the subset of MB neurons (~ 10%) are activated after these trainings (Awata H., et al, PNAS, 2019), and therefore dissociation in this small population is not testable. Accordingly, increase in CoRest-C binding to Rpd3 after these trainings was detectable only using 10 times more flies than that after optogenetic activation, which is mentioned in the legend of Supplementary Fig. 3c.

These results suggest that the neural activation itself triggers the compositional change, regardless of the trainings, and the findings are consistent with our previous findings that both spaced and massed training induced *kay* mRNA expression (Awata H., et al, PNAS, 2019). Although any training can induce activity-dependent gene expression, there are other biological events specifically induced by spaced training, such as the oscillation of MAP kinase activation (Pagani, MR., et al, 2009 Cell), or Arc2 expression in our previous study (Awata H., et al, PNAS, 2019), which allows formation of LTM specifically by spaced training.

3) The major conclusion of this work is that the flexibility of memory updating is already determined in the initial memory consolidation. To strengthen this conclusion, it is necessary to verify that the acute knockdown of CoRest or Rpd3 after memory consolidation does not affect reversal learning.

We thank the reviewer for raising this issue. As we mentioned in the discussion ("The observation that Rpd3/CoRest-F dysfunction increases the flexibility in memory updating is derived from the immediate test after reversal learning, which excludes the effect of

transcription after reversal learning.”), we assumed that the flexibility of memory updating is already determined in the initial memory consolidation. However, as the reviewer mentions, dysfunction of Rpd3/CoRest-F after memory consolidation may be significant enough to increase the flexibility in memory updating. We acutely knocked down *CoRest-F* or *Rpd3* after spaced training, and found that reversal learning was not affected (Supplementary Fig. 12e), which supported our conclusion.

The response to Reviewer #2

In this paper, Takakura and colleagues investigate the role of activity-induced acetylation-mediated changes in the CoRest isoform associated with Rpd3 in memory consolidation and subsequent plasticity. The authors use a series of sophisticated genetic manipulations along with biochemistry and genomic analysis to show that neural activity induces a switch from CoRest-F to CoRest-C association with Rpd3. This switch depends on CBP-mediated acetylation of CoRest-C, and is required for the correct kinetics of repression of activity-induced transcription. Disrupting this pathway prolongs gene activation, and specifically changes the plasticity of memory consolidation. Overall, this is an interesting and complete study, and the data are compelling, rigorous, and well-presented.

Major point:

1) In Figure 1, the rationale for focusing on known Rpd3 binding partners is understandable, but the mass spec data need a bit more explanation. Ideally, there should be some representation of the entirety of the data in the main text (for example a graph showing which Rpd3 associated proteins increase, or at least which known Rpd3 complex members change). If this is too distracting from the main text, it should be presented as a detailed supplementary figure.

We appreciate the reviewer’s suggestion. We added the graph showing the relative amounts of Rpd3 associating proteins, by focusing on Mi-2 (NuRD complex), Sin3A (Sin3 complex) and CoRest (CoRest complex) (Supplementary Fig. 1a). The total amount of all peptides of Mi-2, Sin3A, and CoRest (A-C in Supplementary Table 1) was normalized to that of Rpd3.

Can the authors provide additional explanation about how they selected CoRest from the mass spec data, and whether they think other differences on activation are important?

In this current study, we focused on the known Rpd3-associating proteins, Mi-2, Sin3A, and CoRest. As the reviewer suggests, other proteins may also be important for the function of Rpd3, by acting together with or separately from CoRest, and we just started to analyze other proteins. Regarding this issue, we added the sentence in the results section (line 120-122): “Although other proteins were also found in the Rpd3-immunocomplex, in this study we focused on these known and conserved associating proteins.”

We believe that selection of CoRest out of Mi-2, Sin3A, and CoRest was rational; firstly, neural activation increased the amounts of peptides derived from Mi-2 and CoRest in the Rpd3-immunocomplex, and secondly, knockdown of *CoRest*, but not *Mi-2*, resulted in memory enhancement similarly to that of *Rpd3*.

Also please indicate if the experiment was replicated, and if so how the replicates were handled in the data analysis. The authors should also clarify what they mean by “increased binding” of Mi-2 and CoRest—was this statistically determined, or how was a threshold determined, and which of the metrics indicated in Supplementary Table 1 was used?

We thank the reviewer for raising this issue. Given that the mass-spec analysis was conducted for a screening purpose, the analysis was not replicated, the data were not statistically analyzed, and a threshold was not determined. We therefore revised a sentence to describe this data: “We found the amounts of the peptides derived from Mi-2, a component of the NuRD complex, and CoRest were relatively abundant in the Rpd3-immunocomplex after

thermogenetic activation” (line 117-119). However, we noted that multiple peptides derived from one protein showed similar changes, suggesting that the results of the mass-spec analysis were not an artifact. In addition, CoRest/Rpd3-binding was confirmed through coimmunoprecipitation, and therefore we believe that the main conclusion in this study is solid.

Minor point:

1) In Figure 1, DAPI panels are labelled “Nuclues” please correct to “DNA” or “nuclei” or “DAPI”

We appreciate the reviewer’s comments. We modified the text accordingly.

The response to Reviewer #3

In the study by Mai Takakura et al., the authors sought to understand the basis of the gating mechanism which opens and closes the time window for activity dependent transcription, thought to regulate memory consolidation and if necessary, its update. Specifically, the authors aimed to identify the molecular basis of this mechanism. They focused on the well conserved HDAC2 transcriptional repressor, in *Drosophila* Rpd3, whose activity is involved in both the activity dependent transcription and memory. They embarked on a truly heroic journey starting from a proteomic analysis through biochemical assays to the behavioral validation of their findings. As a result, they found that the gating mechanism of transient transcription involves compositional changes in the Rpd3/CoRest transcriptional repressor complex. Specifically, they determined that the Rpd3/CoRest complex is dissociated upon neural activation by binding of the short CoRest isoform, CoRestC. This compositional change is mediated by acetylation via CPB, and deacetylation via Rpd3. The authors validated those findings in vivo using behavioral assay in *Drosophila*. They determined that Rpd3/CoRest complex has no role in consolidation of the aversive olfactory memory but instead mediates increased flexibility in memory update.

Major comment

This is really a very impressive and exciting work both in terms of approaches and findings. The experiments are carefully controlled, and the conclusions are well supported by the data. In this work the *Drosophila* system was used to its best. The combination of various approaches, from proteomics to behavior, provided an unprecedented detailed picture of the molecular mechanism involved in the activity dependent transcription in memory consolidation and its update.

Minor points

· Both the result and discussion sections of the manuscript are well written. However, the introduction is not very clear and needs some work.

We appreciate the reviewer’s comments. We modified the introduction, especially in the first paragraph, to clarify the background.

· Names of HDAC2, Rpd3, CoRest should not be abbreviated when mentioned for the first time in the text.

We thank the reviewer’s comments. We modified the text accordingly.

· Please explain why kay mRNA is upregulated after ~1.5 hours while Rpd3/CoRest complexes induced by neuronal activation dissociate within 15 minutes?

This is an interesting issue. The compositional change of the Rpd3/CoRest complex was recovered within 15 min. Accordingly, nuclear RNA-seq revealed that the nuclear *kay* mRNA is immediately shut off by 15 min (Fig. 6g), suggesting that our nuclear RNA-seq successfully observed ongoing transcription in the nucleus, without the effect of mRNA in cytosol. We added this explanation in lines 296-299. In contrast, the *kay* mRNA expression after spaced training, which was observed in total RNA, was increased up to 1 hour after the training. Thus, although transcription itself can be immediately shut off, the duration of mRNA expression is affected by the decay kinetics of mRNA. In this regard, we have to consider gene expression as a consequence of multiple mechanisms (transcriptional activation, stability or decay of mRNA), but in current study, we focused on on-going transcription.

- Nucleus is misspelled in DAPI panels of Fig.1a and d
- Degree symbol is messed up in legends of Fig.1 and 5.
- Light is misspelled in Fig.5g (top and bottom).

We appreciate the reviewer's comments. We modified the text accordingly.

Reviewers' Comments:

Reviewer #1:

Remarks to the Author:

All of my concerns have been well addressed.

Reviewer #2:

Remarks to the Author:

The reviewers have carefully addressed the concerns I raised. I have no further concerns about this nice work.

Reviewer #3:

Remarks to the Author:

I would like to reiterate that the study by Mai Takakura et al., is very impressive and exciting both in terms of approaches and findings.

The authors adequately addressed the points I raised. This is an excellent manuscript.